# Anomaly-Gym: A Benchmark for Anomaly Detection in Reinforcement Learning Environments

## Abstract

Anomaly detection (AD) is a key component for deploying reinforcement learning (RL) agents in safety-critical environments, enabling systems to identify unexpected conditions and trigger safe fallback behavior. Despite its importance, research on AD in RL settings is limited. Only a handful of methods have been proposed which - due to the absence of established evaluation scenarios - are evaluated on simple, small-scale, and self-proposed environments. This results in poor comparability and limits systematic analysis of the strengths and weaknesses of current approaches, thus fundamentally impeding progress in this direction. We address this problem by introducing Anomaly-Gym, a comprehensive evaluation suite for AD in RL settings. In contrast to prior work, Anomaly-Gym is based on principled design criteria that disentangle evaluation from methodology. By enforcing specific constraints on the environments and anomalies considered, we propose a broad spectrum of evaluation data that covers both simulated and real-world tasks. In total, our benchmark features 10 different environments, 25 anomaly types, 4 strength levels, as well as multiple sensor modalities. We demonstrate the importance of these different aspects in a series of experiments on pre-generated datasets. For instance, we show that simple methods, while generally neglected in previous work, achieve competitive scores for settings with observational disturbances. In contrast, detecting perturbations of actions or environment dynamics requires more complex methods. Our findings also highlight current challenges with anomaly detection on image data and provide directions for future research.

## 1 Introduction

Anomaly detection (AD) is an essential component of safe and reliable machine learning (ML) systems (Hendrycks et al., 2021). It allows systems to initiate a conservative fallback policy or hand over to human control whenever anomalies are detected that can potentially lead to unsafe or erratic behavior (Nguyen et al., 2015; Amodei et al., 2016). As a long-standing problem in ML, AD has been studied thoroughly in domains such as computer vision (Yang et al., 2024), robotics (Wellhausen et al., 2020), and healthcare (Šabić et al., 2021).

However, the Reinforcement Learning (RL) domain has only witnessed a handful of methods that address AD. The field lacks publicly available benchmark datasets with challenging problems and well-defined evaluation criteria. As a result, existing work is typically evaluated in simpler, small-scale environments, often introduced alongside the corresponding methods. This leads to poor comparability and a limited understanding of the strengths and weaknesses of current approaches, thereby making their practical applicability difficult to assess and hindering progress in the field.

In this work, we address this problem from the bottom up with the following contributions. First, we propose a general framework for evaluating AD within RL settings. Recognizing potential biases in existing evaluation schemes, our framework encompasses a set of principled desiderata and is based on a clear connection to existing literature on AD. Second, we present Anomaly-Gym, a suite of 10 diverse tasks and 25 anomalies designed to rigorously test and compare different aspects of AD for RL under controlled low-level perturbations (see Figure 1).

In contrast to prior work, Anomaly-Gym also incorporates carefully calibrated anomaly strength levels and a real-world setting. We focus on embodied-agent environments because they pose particularly important safety challenges. Third, we demonstrate the utility of Anomaly-Gym in a series of experiments in which we evaluate existing detection methods and baselines across environments, anomaly types, strength levels, and observation modalities. Our results high-light the importance of differentiating between these dimensions, re-

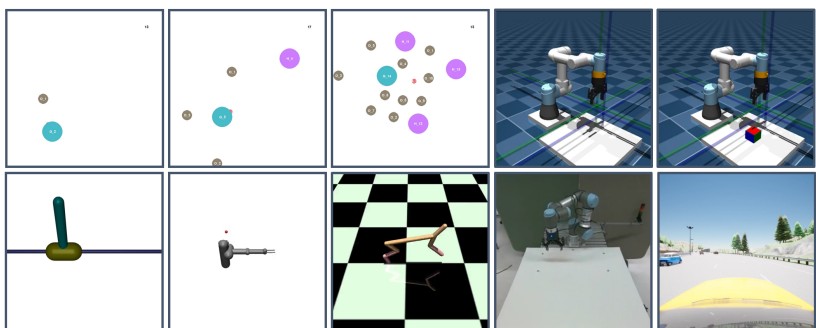

Figure 1: Overview of Anomaly-Gym environments.
*Top*: SAP-Goal{0, 1, 2}, URM-{Reach, Pick and Place (PnP)}
*Bottom*: MJC-{CartpoleSwingup, Reacher3D, HalfCheetah}, URR-Reach, CAR-LaneKeep.

veal the limitations of current approaches on image observations, and show that thresholding detectors on normal data is difficult and materially affects timely anomaly detection.

Anomaly-Gym[1] along with all experiments and datasets[2] is made publicly available.

**Scope of Anomaly-Gym.** Anomaly-Gym is a benchmark for anomaly detection during deployment of embodied RL agents under controlled perturbations to the underlying MDP. Concretely, we focus on low-level observation, action, and dynamics anomalies with configurable onset and calibrated strength, evaluated using external detectors trained on nominal rollouts. This scope is intentionally narrower than the broader problems of robustness, domain generalization, policy adaptation, or semantic novelty in open-world environments. In particular, our goal is not to benchmark every possible distribution shift in RL, but to provide a reproducible and diagnostically useful testbed for detecting deviations that arise from sensor faults, actuator faults, and physical changes in the environment.

## 2   Related Work

**Anomaly Detection in Reinforcement Learning.**   In a relatively recent line of work, several techniques have been proposed that tackle AD specifically in RL settings. See Table 1 for an overview. While several benchmarks exist, they are largely limited to small-scale settings, i.e., simple environments with discrete action spaces and low-dimensional vector observations. As a result, recent methods are often evaluated outside shared benchmarks, and evaluation remains fragmented across self-selected, often non-public environments and datasets. This makes it difficult to compare methods fairly and to disentangle genuine methodological improvements from differences in the evaluation setup.

**Anomaly Detection in other Fields.**   AD has been studied extensively in other fields. Although RL has distinctive characteristics — most notably sequential, policy-dependent, and interactive data generation — many methods from adjacent domains are conceptually relevant and can, in principle, be adapted to RL. **Anomaly detection for i.i.d. data** is a long-standing problem; see Chandola et al. (2009); Chalapathy & Chawla (2019) for surveys of classical and deep learning-based methods. **Time-series anomaly detection** explicitly models temporal dependencies and is therefore particularly relevant to RL; see Lai et al. (2021) for a survey. **Anomaly and OOD detection for image data** is also widely studied, often in classification settings where the goal is to reject samples whose semantics lie outside the support of the training distribution (Yang et al., 2024). **Video anomaly detection** extends this setting to temporal sequences of images and is therefore especially relevant for image-based RL; see Yang et al. (2024); Nayak et al. (2021) for recent surveys.

---

[1]Code: `https://anonymous.4open.science/r/tmlr_anomaly_gym-3C1D`
[2]Datasets: `https://www.kaggle.com/datasets/anonymous31459/anomaly-gym`
both will be made public

Table 1: Related work on AD for RL. **Anomalies**: logic specific to gridworlds (L), action- (A), observation- (O), dynamics- (D) disturbances. **Observation Space**: Vector(V) or Image(I) based. **Action Space**: Discrete or Continuous. **Real:** real-world data. **Pub:** data/code publicly available.

| Ref. | Category | Environments | Anomalies | | | | Obs. | | Act. | | Real | Pub |
|---|---|---|---|---|---|---|---|---|---|---|---|---|
| | | | L | A | O | D | V | I | $\mathbb{N}$ | $\mathbb{R}$ | | |
| Sedlmeier et al. (2019) | Method | Classic Gym, Gridworlds | ✓ | | | | ✓ | | ✓ | | | |
| Sedlmeier et al. (2020) | Method | Classic Gym | | | | ✓ | ✓ | | ✓ | | | |
| Mohammed & Valdenegro (2021) | Benchmark | Classic Gym | | | | ✓ | ✓ | | ✓ | | | |
| Goel et al. (2021) | Benchmark | Gridworlds | ✓ | | | | ✓ | | ✓ | | | ✓ |
| Müller et al. (2022) | Conceptual | | | | | | | | | | | |
| Balloch et al. (2022) | Benchmark | Gridworlds | ✓ | | | | ✓ | | ✓ | | | |
| Danesh & Fern (2021) | Method | Classic Gym, Pybullet Ctrl. | | | ✓ | | ✓ | | | ✓ | | ✓ |
| Haider et al. (2023) | Method | MuJoCo Ctrl. | | ✓ | | ✓ | ✓ | | | ✓ | | ✓ |
| Nasvytis et al. (2024) | Method | Classic Gym | | | | ✓ | ✓ | | | ✓ | | ✓ |
| Martinez et al. (2024) | Analysis | Gridworlds | ✓ | | | | ✓ | | ✓ | | | ✓ |
| Haider et al. (2024) | Analysis | PickAndPlace | | | | ✓ | ✓ | | | ✓ | | |
| Zollicoffer et al. (2024) | Method | Gridworlds | | | ✓ | ✓ | | ✓ | ✓ | | | |
| Zhang et al. (2024) | Method | Classic Gym, Atari, CARLA | | | ✓ | | ✓ | ✓ | ✓ | | | |
| ours | Benchmark | MuJoCo Ctrl., CARLA, Particles, Robot manip. | | ✓ | ✓ | ✓ | ✓ | ✓ | | ✓ | ✓ | ✓ |

In summary, all of the above fields differ in their problem formulation, assumptions, and data characteristics, but they share important features with the problem of AD for RL. Although methods and insights from these areas could in principle be adapted to RL, such transfer has remained limited in practice. We believe this is partly due to the absence of a public and comprehensive evaluation suite. Our work aims to bridge this gap by offering a coherent evaluation framework that is reproducible, comparable, and easy to use, thereby facilitating the transfer of knowledge from related research areas to RL settings. By spanning a broad range of tasks, anomaly types, and observation modalities, Anomaly-Gym enables systematic analysis of existing methods and supports the development of RL-specific anomaly detection approaches.

**Related Benchmark suites.** Beyond anomaly detection, several benchmark suites target robustness and generalization in RL, aiming at the related goal of training robust policies that perform reliably under various disturbances. Procgen Cobbe et al. (2020) evaluates agents under procedural environment generation to test generalization across unseen levels. The Real-World RL suite Dulac-Arnold et al. (2020) emphasizes challenges such as delayed rewards, safety constraints, and stochastic dynamics to assess policy reliability under perturbed conditions. The DMControl generalization benchmark Hansen & Wang (2021) focuses on visual domain shifts (color and background augmentations). However, these suites lack the necessary infrastructure to evaluate anomaly detection. Anomaly-Gym complements these efforts by providing the infrastructure needed for anomaly detection, including controlled and labeled anomaly processes, reproducible data-generation pipelines, standardized evaluation protocols, openly available datasets, reference detector implementations, and interfaces for integrating additional methods (see also Figure 2b).

## 3 Anomaly Detection in Reinforcement Learning

To establish a clear and precise connection between AD and RL contexts, we start with a formal definition of the problem and review the key taxonomy adopted in this work.

### 3.1 Anomaly Detection

**Definition of Anomaly.** *An anomaly is an observation that deviates considerably from some concept of normality* (Chandola et al., 2009).

Following Ruff et al. (2021), this can be formulated more formally via a probabilistic view. Let $\mathcal{X} \subseteq \mathbb{R}^D$ represent the data space associated with a specific task and let $P$ be a probability distribution over $\mathcal{X}$. We define the notion of normality as a probability distribution $P^+$ over $\mathcal{X}$. An anomaly is then an observation $x \in \mathcal{X}$ that resides in a low-probability region under $P^+$ such that

$$\mathbf{A} = \{x \in \mathcal{X} \mid p^+(x) \leq \epsilon\}, \tag{1}$$

where $p^+$ is a pdf of $P$ and $\epsilon \geq 0$ is some threshold.

**Types of Anomalies.** Several anomaly types have been defined in the literature (Chandola et al., 2009). A point anomaly refers to individual anomalous samples $x \in \mathbf{A}$. A group anomaly is a collection of related samples, where the group as a whole exhibits anomalous behavior. A contextual anomaly refers to samples that appear anomalous within a specific context, e.g., time or space.

**Terminology.** While anomaly, outlier, novelty or Out-of-Distribution (OOD) samples are often distinguished, they fundamentally refer to low-probability samples under $P^+$ (Ruff et al., 2021). Consequently, methods for detecting such instances are inherently the same, regardless of the term (OOD, outlier, novelty, anomaly). Therefore, we use the umbrella term *anomaly*.

### 3.2 Connection to Reinforcement Learning

**Reinforcement Learning.** In RL, we consider sequential decision-making problems. Formally, this can be described as a discrete-time Markov Decision Process (MDP) (Puterman, 2014). An MDP is defined by the tuple $\mathcal{M} = (\mathcal{S}, \mathcal{A}, \mathcal{T}, r)$, where $\mathcal{S}$ denotes the state space, $\mathcal{A}$ the action space, $\mathcal{T} : \mathcal{S} \times \mathcal{A} \to \mathcal{S}$ the transition operator that describes the system dynamics, and $r : \mathcal{S} \times \mathcal{A} \to \mathbb{R}$ is the reward function. The RL objective is to find a policy $\pi_\theta : \mathcal{S} \to \mathcal{A}$, parameterized by $\theta$, which selects actions that maximize the expected cumulative sum of future rewards.

**Anomalies in Reinforcement Learning.** The interaction between policy and MDP is the fundamental data-generating process in RL.

*Assumption:* We assume deployment-time anomalies induced by perturbations to the interaction process under a fixed post-training policy; that is, the policy is held constant, and anomalies affect only the interaction process. In particular, we consider perturbations to observations, executed actions, or transition dynamics, but do not consider anomalies that alter the policy itself.

Under this assumption, anomalies can be modeled as perturbations to components of the environment (Haider et al., 2021). Let $\mathcal{M}^+$ denote the nominal environment, and let $\Gamma : \mathcal{M} \to \mathcal{M}$ be a perturbation operator. The corresponding anomalous environment is $\mathcal{M}^- \equiv \Gamma(\mathcal{M}^+)$. For a fixed policy $\pi$, interaction with $\mathcal{M}^+$ and $\mathcal{M}^-$ induces different distributions over samples $x \in \mathcal{X}$. The objective of anomaly detection in RL is therefore to determine whether a sample $x$ is atypical under the nominal deployment distribution $p_{\pi,\mathcal{M}^+}(x)$. Formally, for a threshold $\epsilon > 0$, we define the anomaly set as

$$\mathbf{A} = \{x \in \mathcal{X} \mid p_{\pi,\mathcal{M}^+}(x) \leq \epsilon\}. \tag{2}$$

**Data in Reinforcement Learning.** Data in RL are inherently sequential, contextual, and policy-dependent. Following some policy $\pi_\theta : \mathcal{S} \to \mathcal{A}$ on the MDP $\mathcal{M}$ for $T \in \mathbb{N}$ steps, trajectories are generated:

$$\tau_{\pi,\mathcal{M}} = \{s_0, a_0, s_1, a_1, \ldots, a_{T-1}, s_T\}, \tag{3}$$

where $a_t \sim \pi(\cdot|s_t)$ and $s_{t+1} \sim \mathcal{T}(\cdot|s_t, a_t)$. Consequently, observations, actions, and transitions are temporally correlated and jointly determined by both the policy and the environment. Anomalies in RL must therefore be interpreted relative to this interaction context rather than as isolated i.i.d. samples. This requires consideration of anomalies at different levels: single states $x = s$, individual transitions $x = (s, a, s')$, or entire trajectories $x = \tau$.

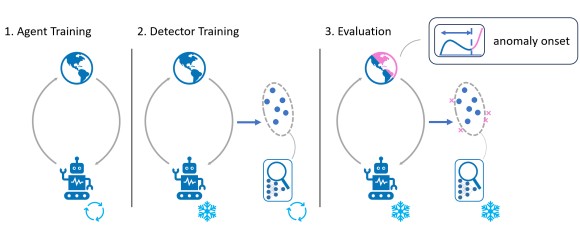
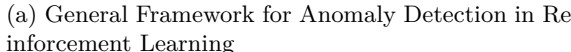

(a) General Framework for Anomaly Detection in Reinforcement Learning

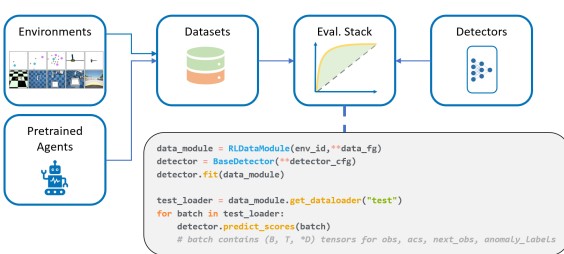

(b) Anomaly Gym infrastructure and interface components

Figure 2: Anomaly-Gym framework and infrastructure

# 4 Evaluating Anomaly Detection in RL

Following the aspects above, we propose a general framework that embeds AD into RL settings.

## 4.1 Framework for Anomaly Detection in Reinforcement Learning

We motivate our evaluation framework from the intended deployment setting. Reinforcement learning agents are typically trained on data generated under nominal conditions in the training environment. During deployment, however, the agent may encounter inputs or dynamics that differ substantially from those observed under nominal operation. Our objective is therefore to monitor the agent–environment interaction at deployment time and identify samples that deviate from the nominal deployment distribution. Such anomaly signals may then be used by a downstream safety mechanism, e.g., to trigger a handover to a human operator or a transition to a safe state.[3] This motivates a three-stage evaluation framework (Figure 2a).

**Stage 1) Agent Training.** The agent interacts with the environment to learn a policy through trial and error. Because the data distribution induced during training changes substantially over the course of learning, especially in early stages, these data are not used to model nominal deployment behavior. At the end of this stage, the policy parameters are fixed.

**Stage 2) Nominal Data Generation and Detector Training.** After training, the fixed policy is rolled out in the nominal MDP to generate deployment-representative data. Anomaly detectors are trained exclusively on this nominal data.

**Stage 3) Evaluation under Deployment Conditions.** The fixed policy is deployed in an environment that may contain anomalies. These anomalies may occur at arbitrary time points during an episode. The goal of the detector is to identify anomalous samples as early and as reliably as possible.

*Extension to adaptive settings.* In settings with periodic online policy updates, the same framework can be applied iteratively. After each policy update, new nominal interaction data can be collected under the updated policy and used to retrain or recalibrate the detector. In this sense, each policy update induces a new nominal deployment distribution.

*Connection to sequential decision-making.* More generally, the framework is policy-agnostic and applies to any controller that induces sequential interaction data, including offline RL policies and classical controllers. In this work, we instantiate it for RL agents. Further discussion is provided in Section 7.

## 4.2 Desiderata towards Evaluation Data

Following our framework from Section 4.1, anomalous data emerge from perturbations of a nominal MDP. Constructing datasets for anomaly detection in RL therefore requires both nominal environments and controlled perturbations that induce anomalous variants of these environments. Below, we define desiderata for

---

[3]This work focuses exclusively on anomaly detection; mitigation strategies are outside the present scope.

each. These criteria are intended to guide the construction of benchmark datasets that support systematic analysis and comparison of anomaly detection methods in RL.

**Environment Desiderata**

- **ED1-Diversity.** Environments should cover a wide range of scenarios and complexities to test the general applicability of AD methods. This includes diverse sensor and actuator modalities.
- **ED2-Scalability.** Environments should cover varying sizes (state/action dimension, horizon length, task complexity) to test the scalability of methods.
- **ED3-Realism.** Environments should incorporate realistic settings to ensure that detection methods are applicable in real-world scenarios. This includes continuous observations and actions.
- **ED4-Solvability.** To ensure that meaningful and non-trivial regions of the state space are reached, RL systems should be able to achieve (partial) success in the considered environments.
- **ED5-Reproducibility.** Environments should allow reproducible evaluation protocols for consistent evaluation results.
- **ED6-Configurability.** Environments should expose parameters that permit controlled modification of relevant aspects of the interaction process.

**Anomaly Desiderata**

- **AD1-Diversity.** Anomalies should encompass different types to broadly evaluate detection capabilities across different failure modalities.
- **AD2-Realism.** Anomalies should mimic realistic faults or unexpected behaviors within the environment to ensure applicability in realistic settings.
- **AD3-Impact.** Anomalies should have varying levels of impact on the environment to evaluate detection capabilities across a spectrum of disruptions.
- **AD4-Difficulty.** Anomalies should be non-trivial to detect, exhibiting characteristics similar to normal operation. For instance, extreme sensor values or shutdown (e.g., a full black/white image) would be trivial to detect.
- **AD5-Calibratability.** Anomalies should admit controlled variation in severity, so that detection methods can be evaluated across graded levels of disruption.

## 5 Anomaly-Gym

We present Anomaly-Gym, a suite of sequential decision-making problems specifically designed to evaluate AD in RL. To satisfy the above-described desiderata, Anomaly-Gym includes and implements the following environments and anomalies. More details are provided in Appendix A.1 and A.2.

### 5.1 Environments

**MuJoCo Control (MJC)** The robotics control tasks from Brockman (2016) serve as widely adopted benchmark for RL algorithms. We include *Cartpole-Swingup*, *Reacher3D*, and *HalfCheetah*.

**Single Agent Particle Env (SAP)** is a set of three simple navigation environments, where the agent controls a particle to reach a goal while avoiding collisions with obstacles. The idea behind this is to mimic existing Grid-world scenarios but with more complex, vector-based (lidar) observations and continuous actions. We developed three levels of difficulty, called *SAP-Goal-{0,1,2}*.

**Universal Robots MuJoCo (URM)** is a set of two different robotic manipulation tasks based on a model of the Universal Robots UR3 in MuJoCo (Todorov et al., 2012). We implemented a simpler *Reach* task and a more complex *Pick-And-Place(PnP)* task.

**Universal Robots Rtde (URR)** is a robotic manipulation environment using a real-world Universal Robots UR3 and an RTDE interface (Lindvig et al., 2025). We implemented a simple *Reach* task with this environment, that mimics the MuJoCo simulation.

**Carla Lanekeep (CAR)** is an autonomous driving environment, where the agent controls a vehicle such that it stays in its lane, keeps a safe distance from other vehicles, and drives at a target speed on a highway. We implemented this environment using CARLA (Dosovitskiy et al., 2017).

## 5.2 Anomalies

**Observation Anomalies** are perturbations to the observations emitted by the environment. We implement the following observation anomalies:

Noise: $o'_t = o_t + \varepsilon_t$     Scaling: $o'_t = o_t \cdot \beta$     Offset: $o'_t = o_t + \beta$

Drift: $o'_t = o_t + \beta * t$     Quantization: $o'_t = \beta \cdot \left\lfloor \frac{o_t}{\beta} \right\rceil$     Temporal Noise: $o'_t = o_t + n_t$

where    $\varepsilon_t \sim \mathcal{N}(0, \beta^2)$,    $n_t = \beta n_{t-1} + \varepsilon_t$, and $\beta \in \mathbb{R}^+$.

In this work, we focus on low-level sensor perturbations and do not include structured visual perturbations such as occlusions. Such perturbations depend strongly on additional factors such as placement, size, or shape, and may introduce semantic and task-dependent label ambiguity. We further discuss this limitation in Section 7.

**Action Anomalies** are perturbations applied to actions before execution in the environment. We consider the same types of perturbations as for observations, except that quantization is replaced by action delay: $a'_t = a_{t-\delta}; \delta \in \mathbb{N}^+$.

**Dynamics Anomalies** refer to perturbations in the underlying dynamics function of an MDP. As they directly affect the transition dynamics, these anomalies are necessarily environment-specific. For example, we implemented moving objects for SAP, changed friction parameters in MJC, disturbance forces in URM.

## 5.3 Anomaly Strengths

Defining anomaly strength in a way that is comparable across anomaly types and environments is non-trivial. For instance, scaling the mass of a robot joint by some factor can lead to an entirely different effect than scaling the policy action by the same factor. Conversely, the exact same anomaly can lead to vastly different results in two different environments. To enable a quantitative comparison of different anomalies in different environments we propose the following process.

Let $J_{\mathcal{M}^+}(\pi_\theta) = \mathbb{E}[\sum_{t=0}^T r_t]$ be the expected return of policy $\pi_\theta$ in the normal MDP, where $r_t$ is the reward received at time step $t$ and $T$ is the time horizon. Let $J_{\mathcal{M}^+}(\pi_R)$ be the expected return of a random policy, and $J_{\mathcal{M}^-}(\pi_\theta)$ the reward in the anomalous MDP. To this end, we define policy degradation through the normalized score

$$\bar{J}_{\mathcal{M}^-}(\pi_\theta) = \frac{J_{\mathcal{M}^-}(\pi_\theta) - J_{\mathcal{M}^+}(\pi_R)}{J_{\mathcal{M}^+}(\pi_\theta) - J_{\mathcal{M}^+}(\pi_R)}. \tag{4}$$

Using this normalized score, we can tune the magnitude of all anomalies and define different strength levels with respect to the degree of degradation of the rollout policy. We set four different levels of anomaly strength, namely *tiny* ($\bar{J}_{\mathcal{M}^-}(\pi_\theta) \approx 0.99$), *medium* (0.9), *strong* (0.75), and *extreme* (0.50).

To determine the corresponding anomaly parameters, we perform a grid search for each anomaly–environment pair and estimate the resulting normalized score from 500 episodes per sample. For some anomaly types, not all strength levels are attainable; such settings are excluded from evaluation. Additional details are provided in Appendix A.4.

## 5.4 Satisfaction of desiderata

In the following we briefly describe how the above described environments and anomalies meet the defined desiderata. Additional details are provided in Appendix A.3

- **ED1-Diversity.** We include 10 environments from four application domains: autonomous driving (CAR), robotic manipulation (URM), control (MJC), and navigation (SAP).

- **ED2-Scalability.** Observation spaces range from $\mathcal{O} \in \mathbb{R}^4$ to $\mathcal{O} \in \mathbb{R}^{26}$ for vector observations, or $\mathbb{R}^{3 \times 256 \times 256}$ for image observations. Action spaces range from $\mathcal{A} \in \mathbb{R}^2$ to $\mathcal{A} \in \mathbb{R}^7$. Episode lengths range from approx. 15 steps in SAP to 500 steps in CAR.

- **ED3-Realism.** The benchmark includes URR, a real-world environment, and CAR, a high-fidelity driving simulator; the remaining environments are based on simulated physics with continuous observation and action spaces representative of deployment-relevant control settings.

- **ED4-Solvability.** For each environment, we train a policy to achieve a success rate above 95% under nominal conditions, whenever possible. Success is defined on a per-environment basis.

- **ED5-Reproducibility.** Evaluation is made reproducible through controlled initial conditions and consistent random seeding.

- **ED6-Configurability.** In simulated environments, all parameters can be adjusted directly. In the real-world environment, only some important parameters can be manipulated directly, while others are approximated through controlled interventions.

- **AD1-Diversity:** We cover three anomaly classes—observation anomalies (sensor faults), action anomalies (actuator faults), and dynamics anomalies (environmental changes)—with 6, 6, and 3 subtypes, for each environment respectively.

- **AD2-Realism:** Observation anomalies mirror real sensor degradation (drift, noise, quantization errors). Action anomalies represent actuator failures (delays, scaling errors, temporally correlated noise). Dynamics anomalies model physical changes (friction degradation, mass shifts, external forces) encountered in real deployments.

- **AD3-Impact:** Our calibration process (Section 5.3) ensures anomalies span four impact levels (tiny to extreme) based on policy degradation, enabling systematic difficulty assessment.

- **AD4-Difficulty:** We explicitly exclude trivial cases (e.g., sensor shutdown, full image corruption). The considered anomalies are designed to preserve plausible system behavior; for example, tiny anomalies induce approx. 1% performance degradation and are therefore non-trivial to detect.

## 5.5 Anomaly-Gym Infrastructure

Beyond the environments and anomaly processes described above, Anomaly-Gym provides the infrastructure required for systematic studies of anomaly detection in RL. Specifically, we release pretrained agents, pre-generated datasets (Section 5.6), efficient data-buffer implementations, evaluation scripts, and unified detector interfaces with reference implementations (Figure 2b). Together, these components lower the barrier to entry and enable more reproducible comparisons.

## 5.6 Pre-Generated Datasets

We release pre-generated datasets collected according to the framework in Section 4.1. For each environment, we first train a TQC policy (Kuznetsov et al., 2020) until it reaches a high task-specific success criterion. For the released datasets, we target a success rate above 95% whenever possible. Because success is environment dependent—for example, maintaining target speed without collision in CAR or placing the object on target in *URM-PnP* — we tune hyperparameters separately for each environment. Additional training details are provided in Appendix A.6.

The final policy is used to generate train and test data for AD. Train data consists of $N$ normal episodes, containing only transitions generated with the normal MDP: $\quad \mathcal{D}_{\text{train}} = \{\tau_{\pi, \mathcal{M}^+}\}_{n=1}^N$.
Test data consists of normal and anomalous episodes: $\quad \mathcal{D}_{\text{test}} = \{\tau_{\pi, \mathcal{M}^+}\}_{n=1}^N \cup \{\tau_{\pi, \mathcal{M}^-}\}_{n=1}^N$.
Anomalous episodes are generated by introducing a perturbation after a randomized number of steps $t_a \in (t_0, t_H)$ (random onset). The timesteps $[t_0, ..., t_{\alpha-1}]$ are labeled as normal, whereas $[t_a, ..., t_H]$ are anomalous. The resulting dataset is balanced in expectation. For *CAR* and *URR*, we collect $N = 50$ episodes in each setting, for all other environments $N = 100$, with different random seeds for each episode.

## 6 Experiments

To assess the utility of the benchmark and the importance of its different design dimensions, we conduct an empirical study using relevant baselines and recent AD methods for RL.

### 6.1 Detection Methods & Baselines

**Selection of detection methods**. The goal of this work is not to provide an exhaustive leaderboard of all possible anomaly-detection methods, but to evaluate a representative set of detector families relevant to RL deployment monitoring. We therefore select methods that cover: (i) classical one-class and nearest-neighbor baselines, which are widely used and provide strong classical reference points; (ii) RL-specific sequence and dynamics-modeling approaches, which can capture action- and transition-dependent anomalies; and (iii) image-based reconstruction, prediction, and representation-based methods, which are relevant for high-dimensional observations. We focus on external, policy-agnostic detectors trained on nominal rollouts, since this matches the benchmark setting and allows methods to be compared independently of the internals of a particular RL algorithm.

For vector observations, we consider the following methods: **IF** Isolation Forest (Liu et al., 2008), **KNN** k-Nearest Neighbor (Cover & Hart, 1967), **OCSVM** One-Class Support Vector Machine (Schölkopf et al., 2001), **RIQN** recurrent implicit quantile networks (Danesh & Fern, 2021), **MLP-DM** dynamics model (DM) with an MLP backbone, **LSTM-DM** DM with an LSTM backbone, and **PE-DM** Probabilistic ensemble DM (Haider et al., 2023).

For image observations we consider: **PredAE** Auto-Encoder predicting the next frame, **KNN-ResNet**, deep-KNN on latent features of a pretrained ResNet, similar to Sun et al. (2022), **PredNet** (Lotter et al., 2017) predicting future frames, **Dino-Patch** nearest-neighbor scoring on DINO patch features, following a simplified variant of (Roth et al., 2022), **ClipSim** Similarity search on CLIP embeddings, similarity search on CLIP embeddings, similar to (Sun et al., 2022; Radford et al., 2021), and **RSSM-PPM**, the Dreamer-style RSSM with PPM score proposed by (Zollicoffer et al., 2024).

Furthermore, we introduce two additional baselines: **LDM** based on a latent DM similar to Haider et al. (2023), but with an image AE and **LRDM**, with an additional Recurrent NN in latent space.

Additional details on all methods are provided in Appendix A.7.

### 6.2 Evaluation Metrics

We evaluate anomaly detection performance using two complementary groups of metrics.

**Aggregate detection metrics.** For aggregate detection performance, we employ established metrics for AD: **AU-ROC**, **AU-PR**, and **FPR95**. For time-series data, these metrics can be computed either **locally** (per sequence, then averaged) or **globally** (pooled over the entire dataset). As we observed no meaningful differences, we report local metrics throughout the main section and defer global metrics to Appendix A.11.

**Timing metrics.** Beyond aggregate discrimination performance, timely anomaly detection is important in sequential decision-making settings. Let $\Delta t = t^* - t_a$ denote the detection delay, where $t_a$ is the ground-truth anomaly onset and $t^*$ is the first time a detector raises an alarm. We consider the following timing metrics:

- **Median Detection Delay (Med Del):** Median value of $\Delta t$ across all sequences.
- **Detection within $j$ steps ($D_j$):** Fraction of anomalies detected within $j \in \{5, 10, 20\}$ steps after onset.
- **Missing Rate (MR):** Fraction of anomalies that are never detected.
- **Early Detection Rate (EDR):** Fraction of detections that occur before the true anomaly onset.

### 6.3 Evaluation

**Overall detection performance.** To assess overall detection performance, we analyze the distribution of AUROC scores across all environments, anomaly types, and strength levels. As shown in Figure 3a,

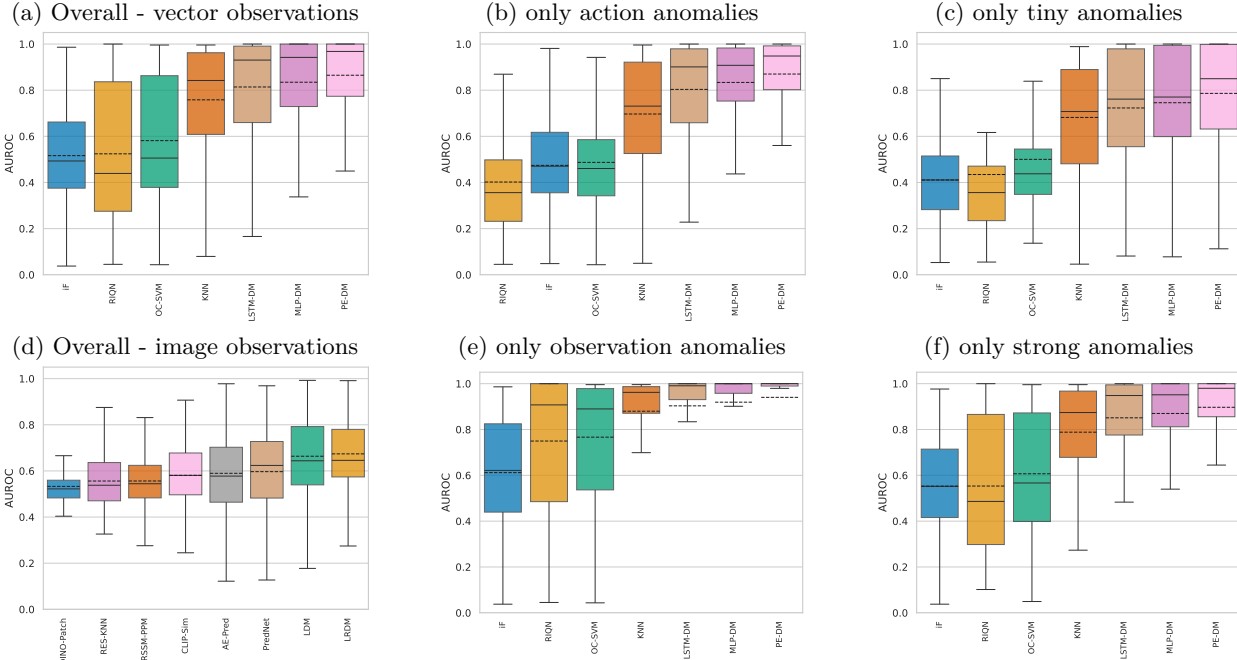

Figure 3: Distribution of AUROC ↑ scores for different detectors. Overall: all environments, anomalies and strengths. Other: all environments but only subgroup of anomaly type/strength. Detectors are ordered from left to right by average AUROC (dashed line).

methods that model environment dynamics (MLP-DM, LSTM-DM, and PE-DM) perform best on vector observations. Among the classical baselines, KNN achieves the strongest performance and even outperforms one NN-based approach.

For image observations, detection performance is generally lower and differences between detectors are smaller (Figure 3d). LRDM achieves the highest scores and outperforms ViT-based approaches. A likely reason is that LRDM learns latent representations from the joint history of observations and actions and thereby captures temporal dynamics, whereas the ViT baselines operate on individual observations and do not explicitly model sequential structure. This indicates that RL-specific inductive biases, particularly temporal dynamics modeling, can improve anomaly detection performance.

**Analysis on Anomaly Types.** Figure 3 b) and e) show detection performance for action and observation anomalies respectively. For observation anomalies, KNN performs comparably with NN-based approaches. In contrast, dynamics-model based approaches clearly outperform other baselines when detecting action anomalies. Unlike observation anomalies, action anomalies are not directly visible and can only be inferred implicitly or through interaction. Overall, these results highlight why anomaly detection in reinforcement learning should be treated as a distinct problem: RL-specific elements—particularly actions and environment dynamics—play a central role in successful detection.

**Analysis on Anomaly Strengths.** Figure 3c) and f) show the influence of anomaly strength on detection performance. In general, we observe that detection performance is strongly correlated with anomaly strength. Interestingly, even the best-performing detectors exhibit high variance for tiny anomalies, whereas this spread is substantially smaller for strong anomalies. This demonstrates that calibration by policy degradation yields meaningful difficulty levels, rather than idiosyncratic parameter choices as common in prior work.

**Environment-level view.** To complement the aggregate analyses above, Table 3 reports the performance of the best detectors across environment groups. In the vector setting, action anomalies are detected best in URR and comparatively well in CAR and MJC at higher strengths, but remain substantially harder in SAP and URM. Dynamics anomalies exhibit even stronger environment dependence, whereas observation

Table 3: AUROC scores for the best performing models (PEDM, LRDM) on tiny / strong anomalies across vector and image observations.

| env. | Vector observations | | | Image observations | | |
|------|--------|----------|-------------|--------|----------|-------------|
|      | action | dynamics | observation | action | dynamics | observation |
| CAR  | 0.87 / 0.99 | 0.51 / 0.65 | 0.81 / 0.97 | 0.71 / 0.76 | 0.53 / 0.53 | 0.65 / 0.69 |
| MJC  | 0.78 / 0.97 | 0.60 / 0.87 | 0.83 / 0.97 | 0.64 / 0.79 | 0.66 / 0.77 | 0.69 / 0.83 |
| SAP  | 0.73 / 0.80 | 0.61 / 0.66 | 0.96 / 0.99 | 0.68 / 0.75 | 0.64 / 0.84 | 0.70 / 0.76 |
| URM  | 0.78 / 0.88 | 0.48 / 0.69 | 0.92 / 0.98 | 0.53 / 0.56 | 0.54 / 0.53 | 0.54 / 0.57 |
| URR  | 0.96 / 0.97 | 0.61 / 0.80 | 0.93 / 0.99 | 0.61 / 0.64 | 0.57 / 0.61 | 0.60 / 0.67 |

Table 4: Detection performance reported as mean $\pm$ standard deviation under different thresholding strategies. ($\downarrow$) lower is better, ($\uparrow$) higher is better.

| Detector | Med. Del ($\downarrow$) | $D_5$ ($\uparrow$) | $D_{10}$ ($\uparrow$) | $D_{20}$ ($\uparrow$) | MR ($\downarrow$) | EDR ($\downarrow$) |
|----------|----------|---------|----------|----------|---------|---------|
| **$3\sigma$ Threshold** | | | | | | |
| KNN     | $16.93_{\pm 25.12}$ | $0.48_{\pm 0.42}$ | $0.53_{\pm 0.41}$ | $0.60_{\pm 0.39}$ | $0.31_{\pm 0.37}$ | $0.34_{\pm 0.31}$ |
| LSTM-DM | $23.37_{\pm 51.82}$ | $0.53_{\pm 0.42}$ | $0.60_{\pm 0.38}$ | $0.67_{\pm 0.34}$ | $0.23_{\pm 0.34}$ | $0.41_{\pm 0.33}$ |
| PE-DM   | $22.26_{\pm 52.39}$ | $0.58_{\pm 0.42}$ | $0.62_{\pm 0.39}$ | $0.68_{\pm 0.35}$ | $0.23_{\pm 0.36}$ | $0.39_{\pm 0.38}$ |
| **95-ID Threshold** | | | | | | |
| KNN     | $10.46_{\pm 17.07}$ | $0.55_{\pm 0.39}$ | $0.61_{\pm 0.37}$ | $0.68_{\pm 0.32}$ | $0.20_{\pm 0.31}$ | $0.54_{\pm 0.34}$ |
| LSTM-DM | $2.51_{\pm 4.82}$ | $0.69_{\pm 0.35}$ | $0.75_{\pm 0.30}$ | $0.78_{\pm 0.26}$ | $0.09_{\pm 0.23}$ | $0.71_{\pm 0.36}$ |
| PE-DM   | $2.88_{\pm 6.01}$ | $0.68_{\pm 0.33}$ | $0.73_{\pm 0.30}$ | $0.77_{\pm 0.26}$ | $0.09_{\pm 0.23}$ | $0.70_{\pm 0.40}$ |
| **Max Threshold** | | | | | | |
| KNN     | $23.46_{\pm 38.45}$ | $0.40_{\pm 0.41}$ | $0.47_{\pm 0.40}$ | $0.54_{\pm 0.40}$ | $0.44_{\pm 0.40}$ | $0.06_{\pm 0.04}$ |
| LSTM-DM | $31.40_{\pm 51.12}$ | $0.44_{\pm 0.42}$ | $0.50_{\pm 0.40}$ | $0.58_{\pm 0.38}$ | $0.48_{\pm 0.42}$ | $0.06_{\pm 0.09}$ |
| PE-DM   | $30.19_{\pm 50.21}$ | $0.46_{\pm 0.43}$ | $0.50_{\pm 0.43}$ | $0.55_{\pm 0.40}$ | $0.45_{\pm 0.42}$ | $0.07_{\pm 0.10}$ |

anomalies are more uniform and nearly saturated at high strength across all environments. In contrast, image-based detection remains far from saturation in every environment group, especially in URM and URR. Overall, the uneven performance across environment groups, strength levels, and anomaly types underscores the need for diverse benchmark coverage, since evaluations on only a limited set of tasks may favor methods that overfit to particular environments rather than generalize across diverse settings.

**Analysis of Detection Timing.** Beyond aggregated detection metrics such as AUROC, the timely detection of anomalies is important. To demonstrate how the benchmark can be used to analyze temporal detection behavior, we evaluate timing-related metrics across detectors at fixed operating points. To this end, we consider three common threshold-selection rules: $3\sigma$, 95id, max (see Appendix A.8 for details). Overall, the results in Table 4 reveal notable differences between detectors in terms of detection latency and reliability. More advanced detectors generally achieve lower median delays and higher detection rates within short horizons, indicating faster and more consistent anomaly identification. At the same time, performance is highly sensitive to the thresholding strategy. Conservative thresholds tend to increase missing rates and detection delays, whereas aggressive thresholds lead to more premature detections. This highlights that threshold calibration remains a challenging aspect of practical anomaly detection, and motivates further investigations of more elaborate thresholding methods.

To complement aggregate metrics, we visualize detection scores over time for two individual episodes with *robot-force* anomalies in the *MJC-Reacher3D* environment. Figure 4 compares *PEDM* and *LRDM* under strong and tiny anomaly settings. In the tiny-anomaly episode, PEDM produces only a small transient response that remains below all thresholds. In the strong-anomaly episode, PEDM reacts sharply at the anomaly onset, with a pronounced score spike that exceeds all thresholds before quickly decaying. In contrast,

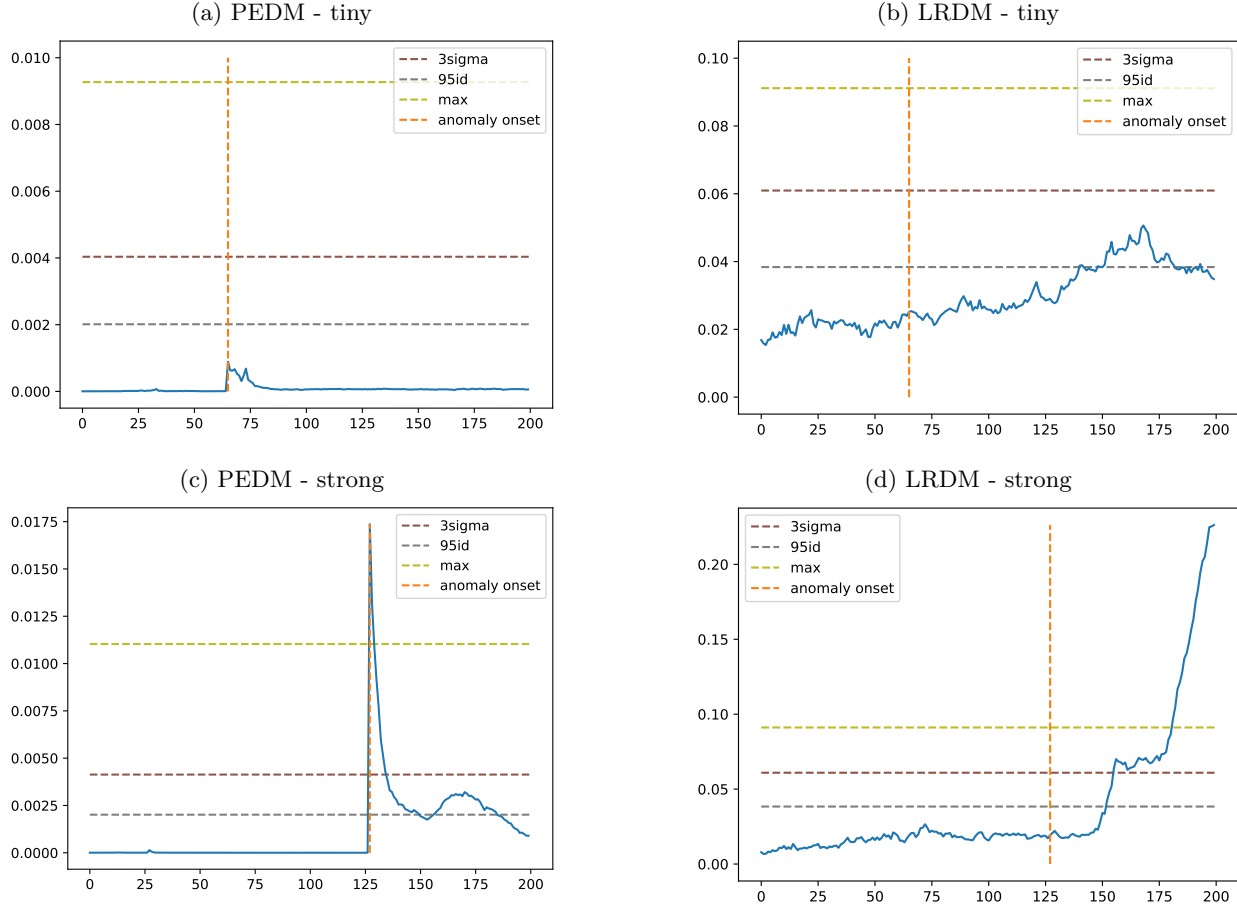

Figure 4: Detection scores over time for two methods, PEDM and LRDM, under strong and tiny anomaly settings for *robot-force* in *MJC-Reacher3D*. The orange vertical dashed line marks the anomaly onset. The horizontal dashed lines indicate different detection thresholds.

Table 5: Influence of Rollout Policy - Comparison of SAC, TD3 and TQC

(a) Norm. Score: strong anomalies

| env | SAC | TD3 | TQC |
|---|---|---|---|
| MJC-Cartpole | 0.894 | 0.893 | 0.967 |
| SAP-Goal0 | 0.948 | 0.897 | 0.957 |
| URM-Reach | 0.840 | 0.810 | 0.940 |

(b) AUROC: Vec. Observations

| detector | SAC | TD3 | TQC |
|---|---|---|---|
| MLP-DM | 0.867 | 0.859 | 0.866 |
| LSTM-DM | 0.899 | 0.902 | 0.887 |
| PE-DM | 0.918 | 0.926 | 0.910 |

(c) AUROC: Img. Observations

| detector | SAC | TD3 | TQC |
|---|---|---|---|
| AE-Pred | 0.634 | 0.655 | 0.613 |
| RES-KNN | 0.659 | 0.657 | 0.623 |
| LRDM | 0.684 | 0.668 | 0.641 |

LRDM shows a more gradual response in these examples. For the tiny anomaly, the score rises moderately and crosses only the 95id threshold while for the strong anomaly, its score increases after a delay and eventually exceeds all thresholds. These observations suggest that PEDM can produce sharp onset-localized responses, whereas LRDM may capture more gradual deviations. This may be attributed to the recurrent component of LRDM. However, these results should be interpreted as illustrative examples rather than general conclusions across all environments and anomaly types.

**Influence of rollout policy.** To study the dependence of detection performance on the rollout policy, we trained two additional RL policies, SAC and TD3, on three environments until they achieved nominal performance comparable to TQC. We then generated additional datasets using these policies while keeping anomaly strength levels fixed to those calibrated with TQC. As shown in Table 5a, policies with similar nominal performance can differ noticeably in their robustness to anomalies, with TQC generally being the

most robust. Table 5b–c further show that, while absolute AUROC depends on the rollout policy, the relative ranking of the strongest detectors remains consistent.

**Sim-to-real alignment**: Trends observed on the real-world URR-Reach task qualitatively match those of its simulated counterpart, URM-Reach (see Appendix A.9). Although validation on additional real-world environments is needed to assess the generality of this result, the finding is encouraging and suggests that insights from Anomaly-Gym may transfer beyond simulation.

## 7 Conclusion, Limitations, and Outlooks

In this work, we introduced Anomaly-Gym, the first unified benchmark suite for AD in RL across multiple domains. Anomaly-Gym offers a diverse set of environments, anomaly types, strength levels, observation modalities, and data from both simulated and real-world tasks. Through a series of experiments, we analyzed the importance of these design dimensions and compared a range of detection methods. Our results show that anomaly type and strength have a substantial impact on detection performance, and they highlight persistent challenges in image-based detection, threshold selection, and timely anomaly detection.

**Limitations (of our evaluations)**
*(i)* We do not consider cross-policy transfer settings, in which a detector is trained on data generated by one policy and evaluated on data generated by another. This is a deliberate design choice intended to reflect the most realistic deployment setting. Nonetheless, Anomaly-Gym supports such evaluations.
*(ii)* We focus on random onset schemes where anomalies persist until the episode ends. Single-point and group events with recovery require further study. Although Anomaly-Gym already technically supports such settings, not all environments permit recovery after failure, which may introduce label ambiguity.
*(iii)* Because there are still few established anomaly-detection methods for image-based RL, we evaluate only a limited set of baselines in that setting. Accordingly, these results should be interpreted as an initial comparison rather than a definitive leaderboard. Future work should expand this evaluation to include a broader range of methods from adjacent areas. At the same time, our findings suggest a promising direction: combining stronger visual encoders, such as ViTs, with models that explicitly capture environment dynamics.
*(iv)* In this work, we treat anomaly detection and robustness as separate problems. In many real-world applications, however, both robustness to minor perturbations and reliable detection of severe anomalies are required. Studying these problems jointly is therefore an important direction for future work, including the question of how detection thresholds should be determined in the presence of policy robustness. Anomaly-Gym provides the tools needed to explore this setting.
*(v)* While the framework introduced in this work is policy agnostic, our experiments focus on data generated by RL policies. This reflects a central motivation for the benchmark: deep RL policies are typically treated as black-box approaches and require external monitoring. This makes AD especially relevant for RL policies. Nonetheless, Anomaly-Gym also supports generating data with arbitrary (e.g. rule-based) policies.

**Outlook (future benchmark extensions)**
*(i)* Anomaly-Gym currently includes one real-world environment. Although the results on this task are consistent with those obtained in simulation, validation across a wider range of real-world applications remains essential. Extending such coverage is challenging, however, because anomalous real-world data are inherently difficult to collect: anomalies are rare and often unsafe to induce, for example due to the risk of hardware damage in robotics or potential harm to subjects in healthcare domains.
*(ii)* We currently omit semantic-visual shifts (e.g. weather changes, novel objects), as well as structured visual perturbations (e.g. occlusions). These shifts are highly environment-specific and may or may not be relevant to the policy. Occlusions are particularly challenging because their severity is not determined only by the amount of visual corruption, but also by additional parameters such as shape, size and placement. This limits cross-task comparability and can potentially introduce label ambiguity. Thus, we believe these topics deserve a dedicated study with carefully designed protocols, which we leave to future work. Anomaly-Gym allows easy implementation of such anomalies. *iii)* Anomaly-Gym currently focuses on single-agent environments, but multi-agent settings are an important direction for future extensions. Conceptually, the framework could be adapted by defining nominal behavior over the joint interaction distribution induced by multiple agents. However, this raises additional challenges, such as non-stationarity or ambiguity in

attributing anomalies to individual agents, their interactions, or the environment. A multi-agent extension would therefore require dedicated environments, anomaly definitions, and evaluation protocols.

Ultimately, Anomaly-Gym provides an important foundation that enables researchers to systematically evaluate and compare novel AD methods and drive the development of robust and reliable RL agents for real-world applications.

## Broader Impact

**Data and Privacy.** The benchmark data are collected in simulated environments or highly controlled robotics settings. Although some tasks use image observations, these capture synthetic or controlled experimental scenes rather than individuals or sensitive real-world activity. The publicly released datasets therefore do not contain personally identifiable information. Consequently, privacy risks are limited.

**Physical Safety.** Introducing perturbations on physical robots can cause unsafe behavior, hardware damage or damage to surroundings. We therefore restrict real-world experiments to supervised, controlled settings in laboratory environments. Perturbations that cannot be introduced safely are instead evaluated in simulation. Extensions to more complex physical systems would require application-specific safeguards and risk assessments.

**Risks and Interpretation of Results.** Despite our efforts to construct a realistic and meaningful testbed, substantial gaps from real-world deployment conditions remain. Benchmark scores from Anomaly-Gym therefore do not constitute evidence that a method is ready for safety-critical deployment. This is important, because detection errors carry substantial consequences: missed anomalies permit unsafe behavior to continue, while false alarms trigger unnecessary shutdowns, disrupt operations and reduce trust in autonomous systems. Real-world use therefore always requires additional application-specific validation, monitoring, and safeguards.

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

# A    Appendix

### A.1    Descriptions of Environments in Anomaly-Gym

See Figure 1 for a visualization of all envs. Table 6 provides a detailed overview of all environments. For even more detail, we refer to the implementation available at: `https://anonymous.4open.science/r/tmlr_anomaly_gym-3C1D`.

### A.2    Descriptions of Anomalies

In the following, we describe all anomalies in more detail.

**Observation Anomalies** are perturbations to the observations emitted by the environment. We implement the following observation anomalies:

- Noise: $o'_t = o_t + e_t$
- Scaling: $o'_t = o_t \cdot \beta$
- Offset: $o'_t = o_t + \beta$
- Drift: $o'_t = o_t + \beta * t$
- Temporal Noise: $o'_t = o_t + \beta\, n_{t-1} + \varepsilon_t +$
- Quantization: $o'_t = \beta \cdot \left\lfloor \frac{o_t}{\beta} \right\rceil$

where     $\varepsilon_t \sim \mathcal{N}(0, \sigma^2), \quad n_0 = \varepsilon_0$ and $\beta \in \mathbb{R}^+$

**Action Anomalies** are perturbations to the actions before they are applied to the environment. We implemented the same types of anomalies as those used for observations but instead of quantization, we add action delay: $a'_t = a_{t-\beta}$ where $\beta \in \mathbb{N}^+$.

**Dynamics Anomalies** are perturbations to the dynamics operator $\mathcal{T} : \mathcal{S} \times \mathcal{A} \to \mathcal{S}$ of the environment. We implement the following dynamics anomalies.

- Body Mass: Body mass is multiplied by a constant factor
- Force Vector: Constant force vector applied to the center of a single robot joint (MJC, URM) or to the center of Agent (SAP)
- Friction: of robot joints changed
- Damping: Inertia of agent is reduced by a factor $\beta$ in each timestep
- Moving Object: Objects moving in uniformly random directions with increasing speed
- Brake Fail: Braking force reduced
- Steer Fail: Steering effect reduced
- Slippery Road: Friction parameters of the wheel to the road surface are reduced
- Moving Goal: Moving Goal back and forth on a straight line
- Control Smoothing: Increased control smoothing of low-level Robot Controllers
- Control Latency: Increased latency of low-level Robot Controllers
- Robot Speed: Max moving speed of robot joints reduced

For more details, we refer to the implementation of each environment's anomalies.

### A.3    Satisfaction of desiderata

Table 6: List of Environments

| Env_id | Description | ObservationTypes/Spaces | Action-Space, Description | Anomaly Types |
|---|---|---|---|---|
| Carla-LaneKeep | Follow the lane at target speed and don't collide with other vehicles | - vector: (9,)
– current speed
– target speed
– current accell.
– current heading
– dist. to lane center
– dist to veh. ahead
– delta vel. to veh. ahead
– last accel.
– last steering angle
- img: (3x256x256) | (2,), acceleration and steering angle | - brake fail
- steer fail
- slippery road
-Action-{Factor, Offset, Noise, Delay, Temp. Noise, Drift}
-Observation-{Factor, Offset, Temp. Noise, Quantization, Drift} |
| MJC-CartpoleSwingup | Swingup the Pole by moving the cart | - vector: (4,)
– car pos.
– cart vel.
– pole angle
– pole vel.
– img: (3x128x128) | (1,) move the cart left/right | - Mass Factor
- Force Vector
- joint friction
-Action-{Factor, Offset, Noise, Delay, Temp. Noise, Drift}
-Observation-{Factor, Offset, Temp. Noise, Quantization, Drift} |
| MJC-Reacher3D | Move Robot EE to goal | - vector: (17,)
– joint pos.
– joint vel.
– img: (3x128x128) | (7,) Torque applied on the robot joints | - Mass Factor
- Force Vector
- joint friction
-Action-{Factor, Offset, Noise, Delay, Temp. Noise, Drift}
-Observation-{Factor, Offset, Temp. Noise, Quantization, Drift}/ |
| MJC-HalfCheetah | Control the HalfCheetah to move as fast as possible | - vector: (18,)
– linear vel.
– joint pos.
– joint vel.
– img: (3x128x128) | (6,) Torque applied on the robot joints | - Mass Factor
- Force Vector
- joint friction
- -Action-{Factor, Offset, Noise, Delay, Temp. Noise, Drift}
-Observation-{Factor, Offset, Temp. Noise, Quantization, Drift} |
| SAP-Goal0 | Move to goal while avoiding collisions with obstacle between agent and goal | - vector: (26,)
– agent pos
– agent vel
– goal pos
– object lidar
– hazard lidar
- img: (3x128x128) | (2,) acceleration in x-/y direction | - Force Agent
- Moving Objects
- Moving Friction
-Action-{Factor, Offset, Noise, Delay, Temp. Noise, Drift}
-Observation-{Factor, Offset, Temp. Noise, Quantization, Drift} |
| SAP-Goal1 | Move to goal while avoiding collisions with multiple obstacles & hazards spawned around the goal | - vector: (26,)
– agent pos
– agent vel
– goal pos
– object lidar
– hazard lidar
- img: (3x128x128) | (2,) acceleration in x-/y direction | - Force Agent
- Moving Objects
- Moving Friction
- -Action-{Factor, Offset, Noise, Delay, Temp. Noise, Drift}
-Observation-{Factor, Offset, Temp. Noise, Quantization, Drift} |
| SAP-Goal2 | Move to goal while avoiding collisions with multiple obstacles & hazards spwaned around the goal | - vector: (26,)
– agent pos
– agent vel
– goal pos
– object lidar
– hazard lidar
- img: (3x128x128) | (2,) acceleration in x-/y direction | - Force Agent
- Moving Objects
- Moving Friction
-Action-{Factor, Offset, Noise, Delay, Temp. Noise, Drift}
-Observation-{Factor, Offset, Temp. Noise, Quantization, Drift} |
| URM-Reach | Move end-effector to target | - vector: (13,)
– ee pos
– ee orientation (rpy)
–goal pos
–goal orientation (quat)
– img: (3x128x128) | (3,) displacement of robot end-effector in cartesian space | - Robot Speed
- Moving Goal
- Robot Friction
-Action-{Factor, Offset, Noise, Delay, Temp. Noise, Drift}
-Observation-{Factor, Offset, Temp. Noise, Quantization, Drift} |
| URM-PnP | Pick up box and move to target | - vector: (13,)
– ee pos
– ee orientation (rpy)
–goal pos
–goal orientation (quat)
–gripper state
–block position
–block orientation
– img: (3x128x128) | (4,) displacement of robot end-effector in cartesian space, gripper distance target | - Robot Speed
- Moving Goal
- Robot Friction
-Action-{Factor, Offset, Noise, Delay, Temp. Noise, Drift}
-Observation-{Factor, Offset, Temp. Noise, Quantization, Drift} |
| URR-Reach | Move end-effector to target | - vector: (13,)
– ee pos
– ee orientation (rpy)
–goal pos
–goal orientation (quat)
– img: (3x256x256) | (3,) displacement of robot end-effector in cartesian space | - Control Latency
- Moving Goal
- Control Smoothing
-Action-{Factor, Offset, Noise, Delay, Temp. Noise, Drift}
-Observation-{Factor, Offset, Temp. Noise, Quantization, Drift} |

Table 7: Satisfaction of Desiderata in anomaly-gym

| desiderata | Satisfaction |
|---|---|
| **ED1-Diversity.** | Anomaly-Gym includes 10 different environments from 4 different fields: Autonomous Driving (CAR), Robotics Manipulation (UR-Envs), Robotics Control (MJC-Control) and navigation (SAP). |
| **ED2-Scalability.** | Environment observation sizes range from $\mathcal{O} \in \mathbb{R}^4$ to $\mathcal{O} \in \mathbb{R}^{26}$ for vector observations or $\mathbb{R}^{3x256x256}$ for image observations. Action spaces range from $\mathcal{A} \in \mathbb{R}^2$ to $\mathcal{A} \in \mathbb{R}^7$ respectively. |
| **ED3-Realism.** | URR is a real world Environment. |
| **ED4-Solvability.** | We train a policy until a success rate of approx. $>95$ % is reached in each environment. Success is defined as follows. |
| | SAP-Goal{1,2,3}: Reach the goal without collision. |
| | CAR-LaneKeep: target speed without collision at end of episode. |
| | URM-PnP: box at target. |
| | MJC-Reacher3D, URM-Reach, URR-Reach: End-effector at target. |
| | Cartpole-Swingup: End-effector displacement smaller than threshold. |
| | MJC-HalfCheetah: Final ground-speed larger than threshold. |
| **ED5-Reproducibility.** | Initial conditions are controlled by consistent random seeding. |
| **ED6-Configurability.** | In simulated environments, all environment parameters can be adjusted directly. In the real-world environment, the most important parameters (for the deployed failure modes) are also controllable. |
| **AD1-Diversity.** | Anomaly-Gym covers a range of 15 anomalies per environment: 6 action anomalies, 6 observation anomalies and 3 environment specific dynamic anomalies, resulting in a total of 25 unique anomaly types (dynmics anomalies are different for each environment). |
| **AD2-Realism.** | Observation anomalies can occur in real world in the form of sensor variations or failure. Action anomalies can occur in the real world in form of actuator variations or failure. Dynamics anomalies can occur in the real world as external influences on the environment/agent. |
| **AD3-Impact.** | We show that all but two anomaly types (moving objects, moving goal) can decrease agent normalized scores by at least as much as 50%. |
| **AD4-Difficulty.** | We exclude anomalies such as sensor/actuator shutdown or failure. All anomalies (especially tiny) minimally alter the original MDP. |

## A.4  Details on anomaly strength tuning

As outlined in Section 5.3, we tune the magnitude of all anomalies and define different strength levels w.r.t. the degree of degradation of the rollout policy. Let

$$J_{\mathcal{M}}(\pi) = \mathbb{E}\left[\sum_{t=0}^{T} r_t\right]$$

be the average cumulative reward of some policy $\pi$ on some MDP $\mathcal{M}$, where $r_t$ is the reward received at time step $t$ and $T$ is the time horizon. Let $J_{\mathcal{M}^+}(\pi_\theta)$, be the average cumulative reward of the trained policy, $J_{\mathcal{M}^+}(\pi_R)$ be the average cumulative reward of a random policy, and $J_{\mathcal{M}^-}(\pi_\theta)$ be the average cumulative reward of the trained policy on the anomalous MDP.

We define policy degradation via the normalized score:

$$\bar{J}_{\mathcal{M}^-}(\pi_\theta) = \frac{J_{\mathcal{M}^-}(\pi_\theta) - J_{\mathcal{M}^+}(\pi_R)}{J_{\mathcal{M}^+}(\pi_\theta) - J_{\mathcal{M}^+}(\pi_R)} \tag{5}$$

and set the different levels of anomaly strength at

- *tiny*: $\bar{J}_{\mathcal{M}^-}(\pi_\theta) \approx 0.99$,

- *medium*: $\bar{J}_{\mathcal{M}^-}(\pi_\theta) \approx 0.90$

- *strong*: $\bar{J}_{\mathcal{M}^-}(\pi_\theta) \approx 0.75$

- *extreme*: $\bar{J}_{\mathcal{M}^-}(\pi_\theta) \approx 0.50$

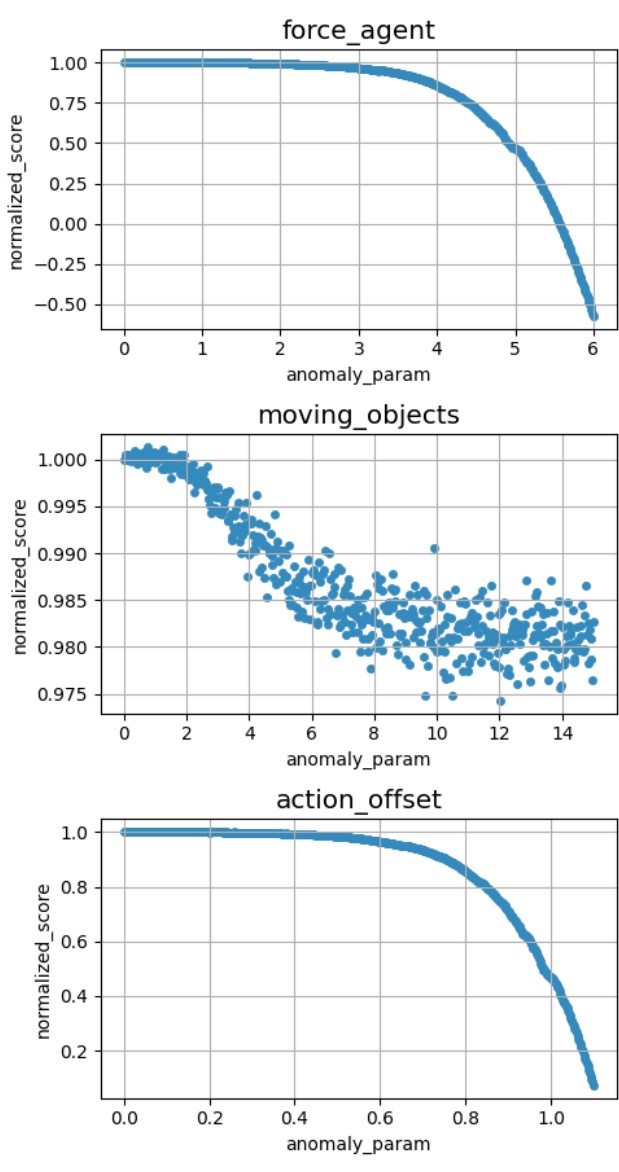

Figure 5: Example of anomaly strength tuning process in SAP-Goal1.

To do this, we ran a fine grid search for each anomaly, strength and environment combination. Since this is a noisy process, we used 500 episodes for each sample. For most anomaly types, there is a strong correlation

between anomaly strength and policy performance. For some anomalies, however, increasing the strength parameter does not lead to any further degradation of the policy after some point. This is exemplified in Figure 5, for three different anomaly types in the SAP-Goal1 environment. A force on the agent, as well as an offset of the action, leads to a decrease in policy performance with increasing anomaly strength. Moving objects with higher speeds, on the other hand, do not continue to influence policy performance after a certain point. The same behavior can be observed for all *moving-object* anomalies in SAP, for the *moving-goal* anomaly in URM-PnP and for some of the *action-delay* parameters. We also had to omit *noise* and *temp-noise* anomalies for URR-Reach due to potential hardware damage caused by oscillating control commands. Table 9 provides an overview of omitted environment-anomaly-strength combinations.

Nonetheless, this process allows us to compare all different types of anomalies along the same strength levels and to compare the same anomaly types across different environments.

Table 9: Excluded anomaly-type/strength combinations per environment.

| Env | Anomaly Type | Strength | Reason |
|---|---|---|---|
| CAR-LaneKeep | action delay | medium | target score not achievable |
| MJC-CartpoleSwingup | action delay | tiny | target score not achievable |
| | | medium | target score not achievable |
| | | strong | target score not achievable |
| MJC-HalfCheetah | action delay | tiny | target score not achievable |
| | | medium | target score not achievable |
| | | strong | target score not achievable |
| MJC-Reacher3D | action delay | extreme | target score not achievable |
| SAP-Goal0 | moving objects | strong | target score not achievable |
| | | extreme | target score not achievable |
| SAP-Goal1 | moving objects | strong | target score not achievable |
| | | extreme | target score not achievable |
| SAP-Goal2 | action delay | tiny | target score not achievable |
| | moving objects | strong | target score not achievable |
| | | extreme | target score not achievable |
| URM-PnP | moving goal | extreme | target score not achievable |
| URM-Reach | – | – | – |
| URR-Reach | action delay | tiny | target score not achievable |
| | action noise | all | hdw. damage / osc. control |
| | action tempnoise | all | hdw. damage / osc. control |
| | obs. noise | all | hdw. damage / osc. control |
| | obs. tempnoise | all | hdw. damage / osc. control |

### A.5 Exact Anomaly Parameters

See Table 10

### A.6 Details on agent training

We train a TQC agent Kuznetsov et al. (2020) for each base environment, using an implementation from Raffin et al. (2021) and hyper-parameters presented in Table 11. Hyper-parameters were selected with a TPE (Tree-structured Parzen Estimator) sweep over learning rate, replay-buffer-size, batch-size, network size, gamma and lr-schedule.

### A.7 Details on detection models

All detectors return anomaly labels:

$$D(x) = \begin{cases} 1 & \text{if } \psi(\cdot) > \vartheta \\ 0 & \text{otherwise} \end{cases}, \tag{6}$$

where 1 represents anomalous samples, and 0 normal samples. The score function and additional hyper-parameters are displayed in Table 11. Hyper-parameters were selected with a coarse grid search over the first 5 parameters for each method in this table.

### A.8 Threshold selection

While overall discriminative performance is an important measure of effective anomaly detectors, the timeliness of detection is also a critical aspect. To quantify this, we analyze the detection delay. We define detection delay as:

$$\Delta t = t^* - t_a \tag{7}$$

where $t_a$ is the ground-truth of anomaly onset and $t^*$ the earliest time a detector identifies an anomaly. A positive $\Delta t$ indicates delayed detection (implies false negatives); a negative value means early detection (implies false positives).

For this, detection methods require a fixed operating point, i.e., a specific threshold, beyond which samples are identified as anomalous. In the absence of labeled anomalies, a common approach is to fit a model to the normal data and select a threshold based on the distribution of anomaly scores on normal inputs of a validation set Aggarwal (2016):

1. Three-sigma rule: $\vartheta = \mu + 3\sigma$, where $\mu$ and $\sigma$ are the mean and standard deviation of scores under normal data

2. Quantile threshold: $\vartheta = Q_{0.95}(d(x)|x \in \mathcal{D}_{\text{val}})$, the 95th percentile of anomaly scores on normal data.

3. Max-validation threshold: $\vartheta = \max_{x \in \mathcal{D}_{\text{val}}} d(x)$, the maximum anomaly score on normal data.

Table 12: Details on detection-model parameters and score functions

| model | parameters | score function |
|-------|-----------|----------------|
| IF | n-estimators: 100 | $\psi(s) = -2^{-\frac{E(h(s))}{2 \cdot \ln(n-1)+\gamma}}$ |
| KNN | k: 1 | $\psi(s) = \|s - x_{k=1}\|_2$ |
| OCSVM | kernel: 'rbf' 
 degree: 3 
 gamma: 'scale' 
 coef: 0.0 
 tol: 0.001m 
 nu: 0.5 | $\psi(x) = \sum_{i=1}^{N} \alpha_i K(x, x_i) - \rho$ |

| riqn | gru-units: 64
quantile-embedding-dim: 128
num-quantile-sample: 64
num-tau-sample: 1
lr: 0.001
train-epochs: 250 | $\psi(s, s') = \|f(s) - s'\|$ |
|---|---|---|
| MLP-DM | network-size: 512-256-128
weight-decay: 0.0001
lr: 0.001
train-epochs: 250 | $\psi(s, a, s') = \|f(s, a) - s'\|_2$ |
| LSTM-DM | hidden-dim: 256
num-layers: 1
fully-connected-dim: 128
weight-decay: 0.0001
lr: 0.001
train-epochs: 250 | $\psi(s, a, s') = \|f(s, a) - s'\|_2$ |
| PE-DM | network-size: 512-256-128
weight-decay: 0.0001
ens-size: 5
n-samples: 1000
lr: 0.001
train-epochs=250 | $\psi(s, a, s') = \frac{1}{B}\{\|f(s, a)^b - s'\|_2\}^B$ |
| DINO-Patch | model name: vit-patch16-224 | $\psi(o) = \max_{p \in \mathcal{P}(o)} \min_{p' \in \mathcal{P}(\mathcal{D})} \|f(p) - f(p')\|_2$ |
| KNN-ResNet | model name: Resnet18
k:1 | $\psi(o, o') = \|f(o) - f(o_{k=1})\|_2$ |
| RSSM-PPM | distribution size: 50
deterministic size: 200
recurrent size: 400
transition size: 100
representation size: 100
embedding size: 200 | $\psi(o, a) = \frac{\sum_i^N |\hat{x}^i_{prior} - \hat{x}^i_{posterior}|}{N}$ |
| CLIP-Sim | model name = ViT-B/32 | $\psi(o, o') = \|f(o) - f(o_{k=1})\|_2$ |
| PredAE | channel-sizes: 32-64-128-256
feature-size: 128
lr: 0.0001
train-epochs: 250 | $\psi(o, o') = \|f(o) - o'\|_2$ |
| PredNet | A-channels: (3, 48, 96, 192)
R-channels: (3, 48, 96, 192)
num-layers: 3
nt: 10
lr: 0.001
train-epochs: 100 | $\psi(o_{[t-10:t]}, o') = \|f(o_{[t-10:t]}) - o'\|_2$ |
| LDM | channel-sizes: 32-64-128-256
feature-size: 128
lr: 0.0005
train-epochs: 250 | $\psi(o, a, o') = \|f(o, a) - o'\|_2$ |

| LRDM | channel-sizes: 32-64-128-256 
 feature-size: 128 
 lr: 0.0005 
 recurrent size: 128, recurrent 
 layers: 2 
 train-epochs: 250 | $\psi(o, a, o') = \|f(o, a) - o'\|_2$ |
|---|---|---|

## A.9 Real world experiments

To validate empirical findings with simulated data, Anomaly-Gym also includes one real-world environment. *URR-Reach* is in its core a replica of its simulated version *URM-Reach*. Instead of relying on the MuJoCo physics simulator Todorov et al. (2012), *URR-Reach* employs a real-time *RTDE*Lindvig et al. (2025) interface to a physical UR3CB robotic manipulator, as well as an Intel RealSense camera interface for obtaining image observations. Apart from these interfaces, both environments are identical. We can thus compare both environments on a 1-to-1 basis.

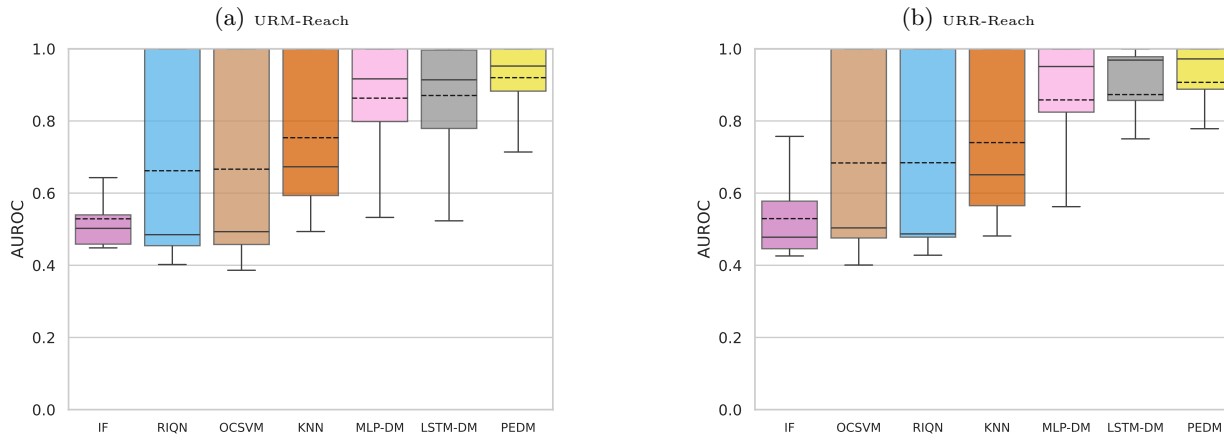

Figure 6: Simulated vs real-world data (vector observations)

To gather data with *URR-Reach*, we use a policy trained in MuJoco and apply it zero-shot to the real-world task. Since real-world interaction is time-consuming and expensive, we only collect 50 episodes for each train/test/val/anomaly setting instead of 100. Figure 6 compares the distribution of AUROC scores of different detectors in both environments. Although there is a visible difference, the general trend and order stay the same. More fine-grained results are provided in Appendix A.11

## A.10 compute resources

We performed all but one of our experiments on a single node of a local compute cluster with the following configuration:

- GPU NVIDIA L40 (46068MiB) - RAM 64GB - CPU AMD EPYC 9334 32-Core

All experiments with vector observations consumed <5GB of GPU Memory peak. Experiments with image observations were more memory-intensive. GPU Memory consumption was, however, still moderate (<10GB), apart from one exception: Experiments with CAR-LaneKeep, which, due to its long episode length, required between 5 GB for Knn-ResNet (the most memory efficient model) and 72GB for PredNet (the most memory extensive model). For the latter, we ran on an NVIDIA H100.

Table 10: Exact Anomaly Parameters

|  | anomaly | medium | strong | tiny | extreme |
|---|---|---|---|---|---|
| CAR-LaneKeep | action scaling | 2.91 | 4.18 | 1.73 | 5.91 |
|  | action noise | 0.86 | 1.04 | 0.31 | 1.41 |
|  | action offset | 0.55 | 0.65 | 0.35 | 0.86 |
|  | brake fail | 0.11 | 0.08 | 0.40 | 0.04 |
|  | observation scaling | 1.41 | 2.06 | 1.00 | 3.61 |
|  | observation noise | 0.09 | 0.17 | 0.01 | 0.32 |
|  | observation offset | 0.04 | 0.05 | 0.01 | 0.31 |
|  | slippery road | 0.11 | 0.09 | 0.22 | 0.06 |
|  | steer fail | 0.24 | 0.10 | 0.32 | 0.05 |
| MJC-CartpoleSwingup | action scaling | 2.19 | 3.21 | 1.38 | 4.58 |
|  | action noise | 1.11 | 1.44 | 0.40 | 1.74 |
|  | action offset | 0.82 | 0.90 | 0.49 | 0.96 |
|  | observation scaling | 1.03 | 1.04 | 1.01 | 1.05 |
|  | observation noise | 0.09 | 0.11 | 0.02 | 0.15 |
|  | observation offset | 0.06 | 0.09 | 0.02 | 0.14 |
|  | robot force | 9.77 | 14.58 | 3.46 | 21.54 |
|  | robot friction | 1.58 | 2.00 | 0.60 | 2.18 |
|  | robot mass | 1.63 | 2.72 | 1.26 | 3.29 |
| MJC-HalfCheetah | action scaling | 1.16 | 1.26 | 1.09 | 1.59 |
|  | action noise | 0.09 | 0.15 | 0.03 | 0.27 |
|  | action offset | 0.08 | 0.11 | 0.03 | 0.16 |
|  | observation scaling | 1.08 | 1.14 | 1.03 | 1.31 |
|  | observation noise | 0.01 | 0.01 | 0.00 | 0.03 |
|  | observation offset | 0.01 | 0.01 | 0.00 | 0.04 |
|  | robot force | 10.38 | 22.44 | 1.19 | 32.55 |
|  | robot friction | 4.21 | 8.96 | 0.48 | 14.79 |
|  | robot mass | 1.13 | 1.21 | 1.02 | 1.40 |
| MJC-Reacher3D | action scaling | 2.28 | 2.50 | 1.36 | 2.50 |
|  | action noise | 0.69 | 1.18 | 0.19 | 1.80 |
|  | action offset | 0.41 | 0.65 | 0.10 | 0.82 |
|  | observation scaling | 1.35 | 1.70 | 1.10 | 2.51 |
|  | observation noise | 0.44 | 1.49 | 0.07 | 5.00 |
|  | observation offset | 0.20 | 0.41 | 0.06 | 0.62 |
|  | robot force | 7.27 | 10.00 | 3.15 | 10.00 |
|  | robot friction | 0.40 | 0.62 | 0.10 | 0.82 |
|  | robot mass | 9.60 | 33.48 | 1.74 | 75.00 |
| SAP-Goal0 | action scaling | 0.22 | 0.11 | 0.62 | 0.03 |
|  | action noise | 1.02 | 2.00 | 0.29 | 4.50 |
|  | action offset | 0.69 | 0.89 | 0.12 | 1.00 |
|  | force agent | 3.45 | 4.46 | 0.60 | 5.01 |
|  | mass agent | 4.46 | 8.92 | 1.61 | 20.00 |
|  | moving objects | 4.87 | - | 0.91 | - |
|  | observation scaling | 2.57 | 4.08 | 1.41 | 9.93 |
|  | observation noise | 0.47 | 0.84 | 0.04 | 1.53 |
|  | observation offset | 0.30 | 0.40 | 0.10 | 0.47 |
| SAP-Goal1 | action scaling | 0.22 | 0.11 | 0.55 | 0.03 |
|  | action noise | 1.55 | 2.40 | 0.67 | 5.00 |
|  | action offset | 0.78 | 0.88 | 0.50 | 0.99 |
|  | force agent | 3.88 | 4.42 | 2.49 | 4.97 |
|  | mass agent | 4.50 | 8.92 | 1.80 | 19.92 |
|  | moving objects | 13.44 | - | 3.47 | - |
|  | observation scaling | 3.22 | 4.88 | 1.63 | 10.00 |
|  | observation noise | 0.51 | 0.79 | 0.22 | 1.53 |
|  | observation offset | 0.26 | 0.36 | 0.13 | 0.43 |
| SAP-Goal2 | action scaling | 0.21 | 0.10 | 0.55 | 0.01 |
|  | action noise | 1.22 | 1.89 | 0.54 | 3.59 |
|  | action offset | 0.67 | 0.84 | 0.37 | 0.95 |
|  | force agent | 3.35 | 4.19 | 1.85 | 4.74 |
|  | mass agent | 4.69 | 9.61 | 1.91 | 20.00 |
|  | moving objects | 34.07 | - | 3.01 | - |
|  | observation scaling | 1.61 | 2.43 | 1.16 | 9.95 |
|  | observation noise | 0.29 | 0.45 | 0.10 | 0.74 |
|  | observation offset | 0.14 | 0.24 | 0.06 | 0.60 |
| URM-Reach | action factor | 0.915 | 0.795 | 0.590 | 0.301 |
|  | action noise | 0.376 | 0.667 | 0.929 | 1.341 |
|  | action offset | 0.295 | 0.501 | 0.657 | 0.794 |
|  | moving goal | 0.003 | 0.004 | 0.004 | 0.004 |
|  | obs factor | 1.032 | 1.087 | 1.095 | 1.107 |
|  | obs noise | 0.002 | 0.006 | 0.016 | 0.026 |
|  | obs offset | 0.018 | 0.023 | 0.024 | 0.026 |
|  | robot friction | 13.026 | 80.160 | 190.381 | 468.938 |
|  | robot speed | 0.009 | 0.003 | 0.003 | 0.002 |
| URR-Reach | action factor | 0.935 | 0.782 | 0.596 | 0.313 |
|  | action noise | 0.263 | 0.566 | 0.990 | 1.394 |
|  | action offset | 0.364 | 0.545 | 0.687 | 0.788 |
|  | control smoothing | 0.061 | 0.077 | 0.104 | 0.173 |
|  | control latency | 0.007 | 0.020 | 0.042 | 0.098 |
|  | moving goal | 0.001 | 0.003 | 0.003 | 0.004 |
|  | obs factor | 1.039 | 1.082 | 1.094 | 1.103 |
|  | obs noise | 0.002 | 0.008 | 0.016 | 0.024 |
|  | obs offset | 0.016 | 0.023 | 0.025 | 0.025 |
| URM-PnP | action factor | 0.871 | 0.709 | 0.491 | 0.305 |
|  | action noise | 0.295 | 0.402 | 0.492 | 0.742 |
|  | action offset | 0.091 | 0.242 | 0.447 | 0.561 |
|  | moving goal | 0.003 | 0.006 | 0.042 | - |
|  | obs factor | 1.227 | 1.439 | 1.712 | 2.091 |
|  | obs noise | 0.002 | 0.005 | 0.008 | 0.011 |
|  | obs offset | 0.006 | 0.012 | 0.017 | 0.022 |
|  | robot friction | 8.838 | 46.717 | 0.001 | 121.212 |
|  | robot speed | 0.004 | 0.003 | 0.002 | 0.001 |

Table 11: Hyper parameters

| Parameter | CAR/SAP | MJC | UR-* |
|---|---|---|---|
| learning rate | 0.0003 | 0.001 | 0.005 |
| replay buffer size | 1e6 | 1e6 | 1e6 |
| learning starts | 100 | 1e3 | 1e3 |
| batch size | 256 | 512 | 512 |
| tau | 0.005 | 0.005 | 0.005 |
| gamma | 0.99 | 0.98 | 0.98 |
| train freq | 1 | 1 | 1 |
| gradient steps | 1 | 1 | 1 |
| top quantiles to drop per net | 2 | 2 | 2 |
| Network size | 256-256 | 512-512-512 | 512-512-512 |
| n critics | 2 | 2 | 2 |
| lr schedule | none | linear anneal. | linear anneal. |

## A.11 Detailed Results

Table 13 and Table 14 provide per-environment results for vector and image observations, respectively. Rows are grouped by environment and anomaly strength, with detectors listed within each group. For each setting, we report mean ± std of AUROC, AUPR, and FPR95, together with the corresponding global metrics (g.). Readers interested in ranking detectors should compare AUROC, AUPR, and FPR95 within each environment and anomaly-strength level. AUROC reflects overall discriminative ability, AUPR emphasizes performance under class imbalance, and FPR95 captures false-positive behavior at a high-recall operating point.

Table 13: Detailed results on vector observations for all detectors and environments. The mean and standard deviation are reported for each detector, environment, and anomaly strength. The results are grouped by environment and detector.

| env | strength | detector | AUROC mean | AUROC std | AUPR mean | AUPR std | FPR95 mean | FPR95 std | g. AUROC mean | g. AUROC std | g. AUPR mean | g. AUPR std | g. FPR95 mean | g. FPR95 std |
|---|---|---|---|---|---|---|---|---|---|---|---|---|---|---|
| CAR-LaneKeep | strong | KNN | 0.91 | 0.14 | 0.89 | 0.14 | 0.21 | 0.24 | 0.92 | 0.13 | 0.91 | 0.14 | 0.25 | 0.33 |
| | | LSTM-DM | 0.89 | 0.18 | 0.87 | 0.18 | 0.25 | 0.32 | 0.90 | 0.17 | 0.89 | 0.18 | 0.28 | 0.38 |
| | | MLP-DM | 0.89 | 0.18 | 0.85 | 0.18 | 0.26 | 0.31 | 0.90 | 0.17 | 0.87 | 0.18 | 0.29 | 0.37 |
| | | OC-SVM | 0.76 | 0.20 | 0.75 | 0.19 | 0.47 | 0.26 | 0.81 | 0.17 | 0.77 | 0.20 | 0.49 | 0.34 |
| | | PE-DM | 0.92 | 0.16 | 0.90 | 0.16 | 0.19 | 0.30 | 0.92 | 0.15 | 0.91 | 0.16 | 0.22 | 0.36 |
| | | RIQN | 0.72 | 0.20 | 0.72 | 0.18 | 0.54 | 0.26 | 0.76 | 0.18 | 0.72 | 0.21 | 0.60 | 0.31 |
| | | iF | 0.71 | 0.19 | 0.70 | 0.17 | 0.55 | 0.19 | 0.76 | 0.17 | 0.71 | 0.19 | 0.56 | 0.28 |
| | tiny | KNN | 0.75 | 0.19 | 0.73 | 0.15 | 0.46 | 0.28 | 0.76 | 0.18 | 0.71 | 0.18 | 0.52 | 0.32 |
| | | LSTM-DM | 0.70 | 0.24 | 0.69 | 0.18 | 0.56 | 0.34 | 0.74 | 0.22 | 0.71 | 0.21 | 0.57 | 0.36 |
| | | MLP-DM | 0.69 | 0.23 | 0.67 | 0.17 | 0.57 | 0.32 | 0.73 | 0.21 | 0.68 | 0.20 | 0.59 | 0.35 |
| | | OC-SVM | 0.43 | 0.09 | 0.49 | 0.06 | 0.76 | 0.09 | 0.52 | 0.09 | 0.46 | 0.07 | 0.85 | 0.14 |
| | | PE-DM | 0.78 | 0.24 | 0.77 | 0.21 | 0.46 | 0.39 | 0.78 | 0.24 | 0.77 | 0.24 | 0.48 | 0.42 |
| | | RIQN | 0.41 | 0.08 | 0.50 | 0.05 | 0.80 | 0.03 | 0.49 | 0.07 | 0.46 | 0.05 | 0.94 | 0.08 |
| | | iF | 0.41 | 0.09 | 0.49 | 0.06 | 0.78 | 0.07 | 0.48 | 0.08 | 0.45 | 0.06 | 0.87 | 0.09 |
| MJC-CartpoleSwingup | strong | KNN | 0.87 | 0.21 | 0.87 | 0.17 | 0.28 | 0.24 | 0.93 | 0.14 | 0.94 | 0.11 | 0.20 | 0.29 |
| | | LSTM-DM | 0.90 | 0.21 | 0.89 | 0.18 | 0.17 | 0.26 | 0.93 | 0.16 | 0.95 | 0.12 | 0.17 | 0.33 |
| | | MLP-DM | 0.91 | 0.21 | 0.91 | 0.17 | 0.13 | 0.25 | 0.94 | 0.16 | 0.96 | 0.11 | 0.15 | 0.34 |
| | | OC-SVM | 0.49 | 0.13 | 0.51 | 0.07 | 0.70 | 0.14 | 0.66 | 0.11 | 0.59 | 0.08 | 0.53 | 0.20 |
| | | PE-DM | 0.94 | 0.16 | 0.93 | 0.16 | 0.13 | 0.26 | 0.96 | 0.12 | 0.97 | 0.09 | 0.14 | 0.31 |
| | | RIQN | 0.50 | 0.14 | 0.50 | 0.07 | 0.79 | 0.20 | 0.65 | 0.12 | 0.57 | 0.07 | 0.68 | 0.28 |
| | | iF | 0.43 | 0.14 | 0.48 | 0.04 | 0.74 | 0.13 | 0.62 | 0.12 | 0.54 | 0.06 | 0.55 | 0.21 |
| | tiny | KNN | 0.74 | 0.30 | 0.76 | 0.19 | 0.47 | 0.26 | 0.83 | 0.23 | 0.83 | 0.19 | 0.35 | 0.32 |
| | | LSTM-DM | 0.77 | 0.30 | 0.78 | 0.22 | 0.32 | 0.32 | 0.83 | 0.26 | 0.82 | 0.22 | 0.31 | 0.37 |
| | | MLP-DM | 0.82 | 0.32 | 0.85 | 0.23 | 0.24 | 0.35 | 0.86 | 0.26 | 0.88 | 0.20 | 0.26 | 0.40 |
| | | OC-SVM | 0.32 | 0.09 | 0.42 | 0.03 | 0.79 | 0.10 | 0.49 | 0.10 | 0.45 | 0.04 | 0.69 | 0.21 |
| | | PE-DM | 0.87 | 0.24 | 0.87 | 0.21 | 0.24 | 0.39 | 0.90 | 0.21 | 0.91 | 0.18 | 0.25 | 0.41 |
| | | RIQN | 0.27 | 0.07 | 0.41 | 0.02 | 0.94 | 0.09 | 0.39 | 0.12 | 0.41 | 0.04 | 0.92 | 0.15 |
| | | iF | 0.29 | 0.12 | 0.43 | 0.02 | 0.82 | 0.11 | 0.47 | 0.13 | 0.46 | 0.03 | 0.67 | 0.20 |
| MJC-HalfCheetah | strong | KNN | 0.92 | 0.04 | 0.92 | 0.04 | 0.32 | 0.18 | 0.96 | 0.02 | 0.96 | 0.02 | 0.17 | 0.13 |
| | | LSTM-DM | 0.90 | 0.06 | 0.91 | 0.06 | 0.37 | 0.24 | 0.94 | 0.04 | 0.94 | 0.03 | 0.24 | 0.20 |
| | | MLP-DM | 0.91 | 0.06 | 0.91 | 0.06 | 0.34 | 0.24 | 0.94 | 0.04 | 0.94 | 0.04 | 0.24 | 0.20 |
| | | OC-SVM | 0.61 | 0.12 | 0.67 | 0.10 | 0.78 | 0.12 | 0.68 | 0.13 | 0.70 | 0.11 | 0.86 | 0.10 |
| | | PE-DM | 0.93 | 0.05 | 0.94 | 0.05 | 0.26 | 0.23 | 0.96 | 0.03 | 0.96 | 0.03 | 0.17 | 0.16 |
| | | RIQN | 0.53 | 0.10 | 0.60 | 0.08 | 0.89 | 0.06 | 0.59 | 0.10 | 0.62 | 0.09 | 0.89 | 0.04 |
| | | iF | 0.61 | 0.09 | 0.63 | 0.08 | 0.76 | 0.10 | 0.72 | 0.07 | 0.68 | 0.08 | 0.59 | 0.10 |
| | tiny | KNN | 0.67 | 0.15 | 0.70 | 0.13 | 0.60 | 0.16 | 0.79 | 0.12 | 0.74 | 0.12 | 0.41 | 0.13 |
| | | LSTM-DM | 0.51 | 0.22 | 0.60 | 0.14 | 0.80 | 0.19 | 0.64 | 0.20 | 0.63 | 0.15 | 0.68 | 0.24 |
| | | MLP-DM | 0.52 | 0.23 | 0.60 | 0.14 | 0.76 | 0.21 | 0.65 | 0.20 | 0.63 | 0.14 | 0.64 | 0.24 |
| | | OC-SVM | 0.44 | 0.05 | 0.50 | 0.03 | 0.74 | 0.07 | 0.49 | 0.08 | 0.49 | 0.04 | 0.87 | 0.04 |
| | | PE-DM | 0.65 | 0.21 | 0.70 | 0.16 | 0.60 | 0.23 | 0.76 | 0.17 | 0.73 | 0.15 | 0.48 | 0.22 |
| | | RIQN | 0.39 | 0.05 | 0.49 | 0.03 | 0.93 | 0.05 | 0.45 | 0.05 | 0.48 | 0.03 | 0.92 | 0.06 |
| | | iF | 0.27 | 0.11 | 0.44 | 0.04 | 0.91 | 0.07 | 0.41 | 0.12 | 0.45 | 0.05 | 0.83 | 0.12 |
| MJC-Reacher3D | strong | KNN | 0.95 | 0.05 | 0.92 | 0.08 | 0.13 | 0.10 | 0.91 | 0.10 | 0.88 | 0.14 | 0.33 | 0.28 |
| | | LSTM-DM | 0.96 | 0.07 | 0.93 | 0.10 | 0.08 | 0.15 | 0.98 | 0.03 | 0.98 | 0.03 | 0.11 | 0.23 |
| | | MLP-DM | 0.96 | 0.07 | 0.92 | 0.10 | 0.08 | 0.15 | 0.98 | 0.03 | 0.97 | 0.03 | 0.12 | 0.24 |

Continued on next page

Table 13: Detailed results on vector observations for all detectors and environments. (continued).

| env | strength | detector | AUROC mean | AUROC std | AUPR mean | AUPR std | FPR95 mean | FPR95 std | g. AUROC mean | g. AUROC std | g. AUPR mean | g. AUPR std | g. FPR95 mean | g. FPR95 std |
|---|---|---|---|---|---|---|---|---|---|---|---|---|---|---|
| SAP-Goal0 | strong | OC-SVM | 0.70 | 0.27 | 0.68 | 0.25 | 0.47 | 0.32 | 0.72 | 0.26 | 0.73 | 0.26 | 0.59 | 0.41 |
| | | PE-DM | 0.97 | 0.06 | 0.95 | 0.09 | 0.07 | 0.15 | 0.99 | 0.03 | 0.99 | 0.02 | 0.10 | 0.23 |
| | | RIQN | 0.56 | 0.29 | 0.56 | 0.22 | 0.61 | 0.31 | 0.62 | 0.28 | 0.61 | 0.26 | 0.68 | 0.36 |
| | | iF | 0.60 | 0.29 | 0.61 | 0.23 | 0.60 | 0.32 | 0.68 | 0.25 | 0.66 | 0.25 | 0.62 | 0.36 |
| | tiny | KNN | 0.88 | 0.07 | 0.82 | 0.10 | 0.24 | 0.12 | 0.69 | 0.15 | 0.61 | 0.15 | 0.65 | 0.27 |
| | | LSTM-DM | 0.74 | 0.27 | 0.70 | 0.22 | 0.39 | 0.36 | 0.79 | 0.22 | 0.74 | 0.22 | 0.44 | 0.40 |
| | | MLP-DM | 0.76 | 0.26 | 0.70 | 0.21 | 0.38 | 0.35 | 0.79 | 0.21 | 0.74 | 0.21 | 0.47 | 0.41 |
| | | OC-SVM | 0.38 | 0.07 | 0.42 | 0.04 | 0.74 | 0.06 | 0.41 | 0.07 | 0.40 | 0.05 | 0.93 | 0.05 |
| | | PE-DM | 0.76 | 0.27 | 0.74 | 0.23 | 0.41 | 0.40 | 0.81 | 0.22 | 0.79 | 0.22 | 0.45 | 0.42 |
| | | RIQN | 0.26 | 0.03 | 0.37 | 0.01 | 0.82 | 0.03 | 0.31 | 0.04 | 0.35 | 0.02 | 0.94 | 0.04 |
| | | iF | 0.26 | 0.09 | 0.38 | 0.03 | 0.86 | 0.07 | 0.35 | 0.08 | 0.37 | 0.04 | 0.94 | 0.05 |
| SAP-Goal1 | strong | KNN | 0.68 | 0.22 | 0.77 | 0.12 | 0.59 | 0.20 | 0.81 | 0.23 | 0.90 | 0.13 | 0.56 | 0.33 |
| | | LSTM-DM | 0.84 | 0.24 | 0.90 | 0.12 | 0.25 | 0.32 | 0.86 | 0.27 | 0.94 | 0.11 | 0.27 | 0.37 |
| | | MLP-DM | 0.89 | 0.18 | 0.93 | 0.10 | 0.19 | 0.28 | 0.89 | 0.22 | 0.96 | 0.08 | 0.24 | 0.35 |
| | | OC-SVM | 0.53 | 0.26 | 0.72 | 0.13 | 0.69 | 0.24 | 0.59 | 0.30 | 0.78 | 0.19 | 0.74 | 0.40 |
| | | PE-DM | 0.93 | 0.17 | 0.95 | 0.08 | 0.13 | 0.25 | 0.94 | 0.15 | 0.97 | 0.05 | 0.14 | 0.25 |
| | | RIQN | 0.44 | 0.33 | 0.68 | 0.18 | 0.73 | 0.38 | 0.60 | 0.29 | 0.77 | 0.20 | 0.70 | 0.46 |
| | | iF | 0.52 | 0.24 | 0.72 | 0.10 | 0.69 | 0.23 | 0.54 | 0.29 | 0.74 | 0.17 | 0.80 | 0.32 |
| | tiny | KNN | 0.51 | 0.23 | 0.63 | 0.16 | 0.81 | 0.12 | 0.65 | 0.19 | 0.62 | 0.22 | 0.84 | 0.13 |
| | | LSTM-DM | 0.72 | 0.23 | 0.76 | 0.19 | 0.50 | 0.38 | 0.74 | 0.22 | 0.72 | 0.23 | 0.60 | 0.44 |
| | | MLP-DM | 0.77 | 0.20 | 0.79 | 0.18 | 0.43 | 0.34 | 0.78 | 0.20 | 0.76 | 0.21 | 0.56 | 0.42 |
| | | OC-SVM | 0.52 | 0.21 | 0.64 | 0.14 | 0.75 | 0.13 | 0.57 | 0.22 | 0.56 | 0.25 | 0.88 | 0.19 |
| | | PE-DM | 0.81 | 0.19 | 0.83 | 0.16 | 0.39 | 0.31 | 0.80 | 0.19 | 0.79 | 0.19 | 0.55 | 0.41 |
| | | RIQN | 0.37 | 0.34 | 0.56 | 0.22 | 0.76 | 0.36 | 0.53 | 0.28 | 0.55 | 0.27 | 0.75 | 0.41 |
| | | iF | 0.52 | 0.10 | 0.63 | 0.06 | 0.71 | 0.06 | 0.53 | 0.10 | 0.49 | 0.11 | 0.92 | 0.09 |
| SAP-Goal2 | strong | KNN | 0.74 | 0.22 | 0.86 | 0.09 | 0.46 | 0.24 | 0.73 | 0.25 | 0.86 | 0.12 | 0.60 | 0.37 |
| | | LSTM-DM | 0.71 | 0.23 | 0.86 | 0.11 | 0.47 | 0.32 | 0.71 | 0.25 | 0.86 | 0.12 | 0.63 | 0.40 |
| | | MLP-DM | 0.81 | 0.23 | 0.90 | 0.09 | 0.34 | 0.32 | 0.81 | 0.25 | 0.91 | 0.10 | 0.44 | 0.37 |
| | | OC-SVM | 0.60 | 0.25 | 0.80 | 0.10 | 0.59 | 0.26 | 0.65 | 0.27 | 0.83 | 0.13 | 0.70 | 0.43 |
| | | PE-DM | 0.82 | 0.19 | 0.90 | 0.09 | 0.33 | 0.29 | 0.83 | 0.22 | 0.91 | 0.10 | 0.40 | 0.34 |
| | | RIQN | 0.44 | 0.31 | 0.71 | 0.16 | 0.73 | 0.37 | 0.59 | 0.28 | 0.79 | 0.15 | 0.70 | 0.46 |
| | | iF | 0.65 | 0.27 | 0.83 | 0.09 | 0.52 | 0.25 | 0.63 | 0.30 | 0.82 | 0.13 | 0.72 | 0.36 |
| | tiny | KNN | 0.74 | 0.12 | 0.80 | 0.09 | 0.51 | 0.11 | 0.72 | 0.15 | 0.72 | 0.18 | 0.76 | 0.23 |
| | | LSTM-DM | 0.75 | 0.17 | 0.81 | 0.13 | 0.46 | 0.28 | 0.75 | 0.16 | 0.74 | 0.17 | 0.64 | 0.37 |
| | | MLP-DM | 0.83 | 0.13 | 0.85 | 0.11 | 0.36 | 0.26 | 0.80 | 0.14 | 0.78 | 0.16 | 0.55 | 0.37 |
| | | OC-SVM | 0.63 | 0.19 | 0.73 | 0.12 | 0.60 | 0.13 | 0.68 | 0.18 | 0.70 | 0.19 | 0.79 | 0.28 |
| | | PE-DM | 0.80 | 0.16 | 0.84 | 0.13 | 0.39 | 0.28 | 0.79 | 0.15 | 0.77 | 0.17 | 0.55 | 0.36 |
| | | RIQN | 0.42 | 0.29 | 0.61 | 0.19 | 0.80 | 0.32 | 0.57 | 0.26 | 0.62 | 0.24 | 0.86 | 0.35 |
| | | iF | 0.70 | 0.13 | 0.78 | 0.08 | 0.51 | 0.13 | 0.69 | 0.14 | 0.71 | 0.14 | 0.81 | 0.22 |
| URM-PnP | strong | KNN | 0.67 | 0.30 | 0.71 | 0.24 | 0.49 | 0.36 | 0.72 | 0.27 | 0.76 | 0.23 | 0.53 | 0.46 |
| | | LSTM-DM | 0.74 | 0.31 | 0.74 | 0.23 | 0.38 | 0.39 | 0.81 | 0.24 | 0.80 | 0.20 | 0.38 | 0.42 |
| | | MLP-DM | 0.76 | 0.31 | 0.77 | 0.23 | 0.35 | 0.38 | 0.83 | 0.23 | 0.82 | 0.19 | 0.36 | 0.42 |
| | | OC-SVM | 0.56 | 0.36 | 0.65 | 0.28 | 0.54 | 0.41 | 0.64 | 0.30 | 0.68 | 0.28 | 0.56 | 0.47 |
| | | PE-DM | 0.81 | 0.30 | 0.82 | 0.23 | 0.30 | 0.40 | 0.86 | 0.23 | 0.88 | 0.18 | 0.32 | 0.43 |
| | | RIQN | 0.58 | 0.36 | 0.66 | 0.29 | 0.55 | 0.47 | 0.65 | 0.30 | 0.68 | 0.28 | 0.58 | 0.48 |
| | | iF | 0.41 | 0.26 | 0.53 | 0.15 | 0.80 | 0.24 | 0.55 | 0.20 | 0.58 | 0.17 | 0.82 | 0.24 |
| | tiny | KNN | 0.57 | 0.35 | 0.66 | 0.27 | 0.53 | 0.39 | 0.62 | 0.33 | 0.66 | 0.29 | 0.57 | 0.48 |
| | | LSTM-DM | 0.59 | 0.37 | 0.65 | 0.26 | 0.51 | 0.42 | 0.67 | 0.31 | 0.68 | 0.27 | 0.50 | 0.43 |
| | | MLP-DM | 0.62 | 0.37 | 0.69 | 0.28 | 0.48 | 0.42 | 0.70 | 0.31 | 0.71 | 0.27 | 0.48 | 0.43 |
| | | OC-SVM | 0.51 | 0.40 | 0.63 | 0.29 | 0.55 | 0.42 | 0.59 | 0.34 | 0.64 | 0.30 | 0.57 | 0.48 |
| | | PE-DM | 0.69 | 0.34 | 0.73 | 0.26 | 0.44 | 0.42 | 0.76 | 0.29 | 0.76 | 0.25 | 0.43 | 0.43 |
| | | RIQN | 0.54 | 0.39 | 0.65 | 0.30 | 0.56 | 0.47 | 0.62 | 0.33 | 0.65 | 0.30 | 0.57 | 0.49 |
| | | iF | 0.23 | 0.16 | 0.43 | 0.07 | 0.90 | 0.14 | 0.39 | 0.12 | 0.44 | 0.08 | 0.93 | 0.08 |
| URM-Reach | strong | KNN | 0.81 | 0.17 | 0.79 | 0.17 | 0.42 | 0.32 | 0.82 | 0.16 | 0.82 | 0.17 | 0.48 | 0.42 |
| | | LSTM-DM | 0.95 | 0.08 | 0.90 | 0.09 | 0.14 | 0.19 | 0.95 | 0.08 | 0.93 | 0.09 | 0.17 | 0.25 |
| | | MLP-DM | 0.94 | 0.09 | 0.90 | 0.11 | 0.17 | 0.22 | 0.95 | 0.09 | 0.93 | 0.09 | 0.21 | 0.29 |
| | | OC-SVM | 0.70 | 0.26 | 0.72 | 0.23 | 0.49 | 0.38 | 0.71 | 0.25 | 0.71 | 0.25 | 0.55 | 0.46 |
| | | PE-DM | 0.96 | 0.06 | 0.92 | 0.09 | 0.13 | 0.17 | 0.96 | 0.06 | 0.95 | 0.07 | 0.14 | 0.19 |
| | | RIQN | 0.69 | 0.27 | 0.73 | 0.23 | 0.48 | 0.41 | 0.69 | 0.27 | 0.70 | 0.26 | 0.55 | 0.47 |
| | | iF | 0.57 | 0.10 | 0.59 | 0.06 | 0.75 | 0.09 | 0.59 | 0.10 | 0.58 | 0.09 | 0.85 | 0.08 |
| | tiny | KNN | 0.73 | 0.22 | 0.74 | 0.21 | 0.48 | 0.36 | 0.74 | 0.22 | 0.74 | 0.22 | 0.55 | 0.46 |
| | | LSTM-DM | 0.90 | 0.14 | 0.85 | 0.14 | 0.24 | 0.30 | 0.89 | 0.15 | 0.87 | 0.15 | 0.30 | 0.37 |
| | | MLP-DM | 0.87 | 0.17 | 0.85 | 0.16 | 0.28 | 0.34 | 0.88 | 0.17 | 0.87 | 0.16 | 0.32 | 0.40 |
| | | OC-SVM | 0.68 | 0.26 | 0.71 | 0.23 | 0.50 | 0.38 | 0.69 | 0.26 | 0.70 | 0.26 | 0.56 | 0.47 |
| | | PE-DM | 0.89 | 0.15 | 0.86 | 0.15 | 0.26 | 0.32 | 0.89 | 0.16 | 0.88 | 0.16 | 0.28 | 0.37 |
| | | RIQN | 0.68 | 0.27 | 0.73 | 0.23 | 0.49 | 0.41 | 0.69 | 0.26 | 0.70 | 0.26 | 0.57 | 0.48 |
| | | iF | 0.51 | 0.06 | 0.56 | 0.04 | 0.80 | 0.07 | 0.53 | 0.05 | 0.53 | 0.05 | 0.91 | 0.04 |
| URR-Reach | strong | KNN | 0.76 | 0.19 | 0.74 | 0.20 | 0.49 | 0.34 | 0.79 | 0.18 | 0.77 | 0.20 | 0.52 | 0.41 |
| | | LSTM-DM | 0.91 | 0.14 | 0.84 | 0.14 | 0.24 | 0.30 | 0.91 | 0.14 | 0.88 | 0.14 | 0.27 | 0.36 |
| | | MLP-DM | 0.90 | 0.15 | 0.84 | 0.16 | 0.26 | 0.32 | 0.90 | 0.15 | 0.88 | 0.16 | 0.29 | 0.37 |
| | | OC-SVM | 0.68 | 0.26 | 0.69 | 0.24 | 0.53 | 0.38 | 0.68 | 0.27 | 0.68 | 0.27 | 0.59 | 0.47 |
| | | PE-DM | 0.93 | 0.09 | 0.86 | 0.14 | 0.17 | 0.20 | 0.93 | 0.11 | 0.89 | 0.14 | 0.19 | 0.24 |
| | | RIQN | 0.65 | 0.28 | 0.68 | 0.26 | 0.52 | 0.41 | 0.64 | 0.29 | 0.64 | 0.29 | 0.60 | 0.48 |
| | | iF | 0.57 | 0.13 | 0.57 | 0.09 | 0.75 | 0.11 | 0.58 | 0.13 | 0.57 | 0.14 | 0.89 | 0.10 |
| | tiny | KNN | 0.69 | 0.25 | 0.71 | 0.24 | 0.53 | 0.40 | 0.71 | 0.25 | 0.70 | 0.26 | 0.55 | 0.47 |
| | | LSTM-DM | 0.84 | 0.20 | 0.79 | 0.19 | 0.33 | 0.39 | 0.83 | 0.22 | 0.81 | 0.21 | 0.39 | 0.43 |
| | | MLP-DM | 0.83 | 0.21 | 0.80 | 0.21 | 0.36 | 0.40 | 0.83 | 0.22 | 0.82 | 0.22 | 0.41 | 0.44 |
| | | OC-SVM | 0.68 | 0.26 | 0.70 | 0.24 | 0.50 | 0.39 | 0.67 | 0.28 | 0.67 | 0.28 | 0.57 | 0.49 |
| | | PE-DM | 0.87 | 0.18 | 0.83 | 0.19 | 0.30 | 0.37 | 0.85 | 0.20 | 0.84 | 0.21 | 0.34 | 0.42 |
| | | RIQN | 0.67 | 0.29 | 0.70 | 0.26 | 0.50 | 0.43 | 0.65 | 0.30 | 0.66 | 0.29 | 0.58 | 0.50 |
| | | iF | 0.49 | 0.06 | 0.52 | 0.04 | 0.81 | 0.05 | 0.49 | 0.06 | 0.48 | 0.05 | 0.94 | 0.03 |

Table 14: Detailed results on image observations for all detectors and environments. The mean and standard deviation are reported for each detector, environment, and anomaly strength. The results are grouped by environment and detector.

| env | strength | detector | AUROC | | AUPR | | FPR95 | | g. AUROC | | g. AUPR | | g. FPR95 | |
|---|---|---|---|---|---|---|---|---|---|---|---|---|---|---|
| | | | mean | std | mean | std | mean | std | mean | std | mean | std | mean | std |
| CAR-LaneKeep | strong | AE-Pred | 0.49 | 0.03 | 0.51 | 0.02 | 0.83 | 0.03 | 0.49 | 0.03 | 0.46 | 0.04 | 0.98 | 0.01 |
| | | CLIP-Sim | 0.49 | 0.06 | 0.49 | 0.04 | 0.88 | 0.04 | 0.52 | 0.06 | 0.44 | 0.05 | 0.90 | 0.05 |
| | | DINO-Patch | 0.51 | 0.07 | 0.52 | 0.05 | 0.86 | 0.04 | 0.53 | 0.07 | 0.47 | 0.07 | 0.90 | 0.05 |
| | | LDM | 0.69 | 0.10 | 0.65 | 0.08 | 0.75 | 0.12 | 0.67 | 0.09 | 0.60 | 0.10 | 0.88 | 0.14 |
| | | LRDM | 0.69 | 0.10 | 0.65 | 0.08 | 0.75 | 0.12 | 0.67 | 0.09 | 0.60 | 0.10 | 0.88 | 0.14 |
| | | PredNet | 0.52 | 0.03 | 0.52 | 0.03 | 0.80 | 0.02 | 0.54 | 0.03 | 0.47 | 0.04 | 0.93 | 0.01 |
| | | RES-KNN | 0.47 | 0.06 | 0.50 | 0.04 | 0.81 | 0.04 | 0.52 | 0.05 | 0.44 | 0.05 | 0.88 | 0.05 |
| | | RSSM-PPM | 0.81 | 0.23 | 0.79 | 0.19 | 0.43 | 0.35 | 0.83 | 0.21 | 0.79 | 0.22 | 0.41 | 0.39 |
| | tiny | AE-Pred | 0.47 | 0.01 | 0.53 | 0.01 | 0.84 | 0.01 | 0.48 | 0.01 | 0.47 | 0.02 | 0.98 | 0.01 |
| | | CLIP-Sim | 0.41 | 0.02 | 0.47 | 0.01 | 0.91 | 0.01 | 0.44 | 0.01 | 0.42 | 0.01 | 0.95 | 0.01 |
| | | DINO-Patch | 0.42 | 0.01 | 0.49 | 0.01 | 0.90 | 0.01 | 0.44 | 0.01 | 0.43 | 0.01 | 0.95 | 0.01 |
| | | LDM | 0.65 | 0.11 | 0.61 | 0.07 | 0.74 | 0.13 | 0.63 | 0.10 | 0.57 | 0.08 | 0.85 | 0.14 |
| | | LRDM | 0.65 | 0.11 | 0.61 | 0.07 | 0.74 | 0.13 | 0.63 | 0.10 | 0.57 | 0.08 | 0.85 | 0.14 |
| | | PredNet | 0.49 | 0.01 | 0.53 | 0.01 | 0.82 | 0.01 | 0.52 | 0.01 | 0.48 | 0.02 | 0.93 | 0.01 |
| | | RES-KNN | 0.39 | 0.02 | 0.48 | 0.01 | 0.87 | 0.02 | 0.44 | 0.02 | 0.43 | 0.01 | 0.95 | 0.01 |
| | | RSSM-PPM | 0.61 | 0.23 | 0.64 | 0.17 | 0.69 | 0.26 | 0.65 | 0.23 | 0.62 | 0.21 | 0.70 | 0.31 |
| MJC-CartpoleSwingup | strong | AE-Pred | 0.78 | 0.17 | 0.77 | 0.15 | 0.61 | 0.29 | 0.82 | 0.16 | 0.86 | 0.14 | 0.64 | 0.37 |
| | | CLIP-Sim | 0.66 | 0.13 | 0.64 | 0.11 | 0.81 | 0.15 | 0.68 | 0.15 | 0.70 | 0.13 | 0.86 | 0.15 |
| | | DINO-Patch | 0.49 | 0.06 | 0.52 | 0.04 | 0.90 | 0.09 | 0.53 | 0.09 | 0.54 | 0.06 | 0.94 | 0.05 |
| | | LDM | 0.86 | 0.15 | 0.83 | 0.15 | 0.42 | 0.31 | 0.89 | 0.12 | 0.92 | 0.10 | 0.44 | 0.40 |
| | | LRDM | 0.86 | 0.15 | 0.83 | 0.15 | 0.42 | 0.31 | 0.89 | 0.12 | 0.92 | 0.10 | 0.44 | 0.40 |
| | | PredNet | 0.68 | 0.09 | 0.61 | 0.08 | 0.66 | 0.07 | 0.67 | 0.11 | 0.66 | 0.12 | 0.83 | 0.11 |
| | | RES-KNN | 0.77 | 0.12 | 0.73 | 0.10 | 0.69 | 0.19 | 0.77 | 0.13 | 0.80 | 0.12 | 0.79 | 0.17 |
| | | RSSM-PPM | 0.62 | 0.27 | 0.69 | 0.18 | 0.73 | 0.23 | 0.69 | 0.26 | 0.73 | 0.21 | 0.67 | 0.32 |
| | tiny | AE-Pred | 0.48 | 0.18 | 0.54 | 0.09 | 0.86 | 0.14 | 0.49 | 0.21 | 0.55 | 0.12 | 0.90 | 0.18 |
| | | CLIP-Sim | 0.41 | 0.16 | 0.50 | 0.08 | 0.90 | 0.12 | 0.37 | 0.17 | 0.45 | 0.08 | 0.97 | 0.05 |
| | | DINO-Patch | 0.41 | 0.12 | 0.49 | 0.05 | 0.90 | 0.10 | 0.40 | 0.16 | 0.47 | 0.09 | 0.96 | 0.06 |
| | | LDM | 0.56 | 0.20 | 0.59 | 0.13 | 0.83 | 0.15 | 0.58 | 0.22 | 0.62 | 0.15 | 0.88 | 0.20 |
| | | LRDM | 0.56 | 0.20 | 0.59 | 0.13 | 0.83 | 0.15 | 0.58 | 0.22 | 0.62 | 0.15 | 0.88 | 0.20 |
| | | PredNet | 0.64 | 0.07 | 0.55 | 0.06 | 0.61 | 0.07 | 0.61 | 0.09 | 0.57 | 0.10 | 0.77 | 0.03 |
| | | RES-KNN | 0.57 | 0.17 | 0.57 | 0.10 | 0.81 | 0.18 | 0.50 | 0.18 | 0.54 | 0.12 | 0.91 | 0.10 |
| | | RSSM-PPM | 0.40 | 0.26 | 0.53 | 0.18 | 0.90 | 0.15 | 0.50 | 0.24 | 0.54 | 0.21 | 0.85 | 0.20 |
| MJC-HalfCheetah | strong | AE-Pred | 0.42 | 0.02 | 0.50 | 0.03 | 0.98 | 0.01 | 0.43 | 0.02 | 0.48 | 0.03 | 0.99 | 0.01 |
| | | CLIP-Sim | 0.53 | 0.03 | 0.55 | 0.03 | 0.94 | 0.01 | 0.54 | 0.02 | 0.55 | 0.03 | 0.95 | 0.01 |
| | | DINO-Patch | 0.53 | 0.03 | 0.57 | 0.03 | 0.92 | 0.01 | 0.53 | 0.03 | 0.57 | 0.04 | 0.94 | 0.01 |
| | | LDM | 0.44 | 0.02 | 0.50 | 0.03 | 0.98 | 0.01 | 0.44 | 0.02 | 0.49 | 0.03 | 0.99 | 0.01 |
| | | LRDM | 0.62 | 0.01 | 0.62 | 0.03 | 0.83 | 0.03 | 0.59 | 0.01 | 0.64 | 0.03 | 0.91 | 0.02 |
| | | PredNet | 0.80 | 0.06 | 0.71 | 0.07 | 0.45 | 0.10 | 0.57 | 0.06 | 0.54 | 0.06 | 0.80 | 0.08 |
| | | RES-KNN | 0.58 | 0.06 | 0.62 | 0.06 | 0.92 | 0.02 | 0.61 | 0.05 | 0.66 | 0.06 | 0.92 | 0.02 |
| | | RSSM-PPM | 0.55 | 0.04 | 0.61 | 0.05 | 0.94 | 0.01 | 0.58 | 0.03 | 0.64 | 0.04 | 0.94 | 0.01 |
| | tiny | AE-Pred | 0.45 | 0.01 | 0.52 | 0.01 | 0.97 | 0.01 | 0.46 | 0.01 | 0.51 | 0.01 | 0.97 | 0.00 |
| | | CLIP-Sim | 0.45 | 0.01 | 0.51 | 0.01 | 0.96 | 0.00 | 0.46 | 0.01 | 0.50 | 0.01 | 0.97 | 0.01 |
| | | DINO-Patch | 0.49 | 0.01 | 0.54 | 0.01 | 0.94 | 0.01 | 0.49 | 0.01 | 0.53 | 0.01 | 0.95 | 0.01 |
| | | LDM | 0.47 | 0.01 | 0.52 | 0.01 | 0.96 | 0.01 | 0.47 | 0.01 | 0.51 | 0.01 | 0.97 | 0.01 |
| | | LRDM | 0.59 | 0.01 | 0.56 | 0.02 | 0.73 | 0.02 | 0.56 | 0.01 | 0.56 | 0.02 | 0.83 | 0.03 |
| | | PredNet | 0.96 | 0.00 | 0.90 | 0.02 | 0.16 | 0.01 | 0.82 | 0.02 | 0.80 | 0.03 | 0.57 | 0.03 |
| | | RES-KNN | 0.42 | 0.02 | 0.49 | 0.01 | 0.97 | 0.01 | 0.44 | 0.02 | 0.48 | 0.01 | 0.97 | 0.01 |
| | | RSSM-PPM | 0.39 | 0.01 | 0.49 | 0.01 | 0.97 | 0.01 | 0.43 | 0.01 | 0.48 | 0.01 | 0.97 | 0.00 |
| MJC-Reacher3D | strong | AE-Pred | 0.52 | 0.09 | 0.54 | 0.06 | 0.71 | 0.02 | 0.55 | 0.11 | 0.57 | 0.12 | 0.92 | 0.03 |
| | | CLIP-Sim | 0.79 | 0.07 | 0.69 | 0.08 | 0.57 | 0.08 | 0.77 | 0.10 | 0.76 | 0.12 | 0.75 | 0.09 |
| | | DINO-Patch | 0.79 | 0.07 | 0.70 | 0.07 | 0.53 | 0.06 | 0.79 | 0.09 | 0.77 | 0.12 | 0.70 | 0.10 |
| | | LDM | 0.94 | 0.04 | 0.89 | 0.07 | 0.19 | 0.10 | 0.90 | 0.06 | 0.91 | 0.06 | 0.45 | 0.18 |
| | | LRDM | 0.91 | 0.05 | 0.85 | 0.08 | 0.27 | 0.11 | 0.88 | 0.07 | 0.88 | 0.08 | 0.53 | 0.18 |
| | | PredNet | 0.58 | 0.11 | 0.60 | 0.07 | 0.68 | 0.12 | 0.56 | 0.12 | 0.58 | 0.10 | 0.93 | 0.08 |
| | | RES-KNN | 0.79 | 0.07 | 0.69 | 0.08 | 0.53 | 0.07 | 0.78 | 0.09 | 0.75 | 0.12 | 0.72 | 0.09 |
| | | RSSM-PPM | 0.68 | 0.10 | 0.63 | 0.10 | 0.79 | 0.10 | 0.70 | 0.10 | 0.69 | 0.12 | 0.82 | 0.09 |
| | tiny | AE-Pred | 0.44 | 0.03 | 0.50 | 0.02 | 0.71 | 0.01 | 0.45 | 0.03 | 0.46 | 0.03 | 0.94 | 0.01 |
| | | CLIP-Sim | 0.69 | 0.01 | 0.61 | 0.01 | 0.67 | 0.02 | 0.63 | 0.02 | 0.58 | 0.03 | 0.84 | 0.02 |
| | | DINO-Patch | 0.69 | 0.01 | 0.61 | 0.01 | 0.60 | 0.01 | 0.65 | 0.02 | 0.59 | 0.03 | 0.81 | 0.02 |
| | | LDM | 0.86 | 0.03 | 0.77 | 0.06 | 0.29 | 0.02 | 0.74 | 0.06 | 0.68 | 0.09 | 0.64 | 0.03 |
| | | LRDM | 0.81 | 0.04 | 0.73 | 0.06 | 0.40 | 0.02 | 0.70 | 0.05 | 0.65 | 0.08 | 0.72 | 0.03 |
| | | PredNet | 0.62 | 0.02 | 0.61 | 0.02 | 0.60 | 0.01 | 0.60 | 0.03 | 0.57 | 0.03 | 0.87 | 0.04 |
| | | RES-KNN | 0.69 | 0.02 | 0.60 | 0.02 | 0.64 | 0.02 | 0.62 | 0.03 | 0.55 | 0.03 | 0.82 | 0.02 |
| | | RSSM-PPM | 0.56 | 0.06 | 0.53 | 0.06 | 0.89 | 0.04 | 0.56 | 0.07 | 0.53 | 0.08 | 0.92 | 0.04 |
| SAP-Goal0 | strong | AE-Pred | 0.69 | 0.07 | 0.76 | 0.07 | 0.67 | 0.11 | 0.65 | 0.04 | 0.78 | 0.10 | 0.87 | 0.04 |
| | | CLIP-Sim | 0.72 | 0.03 | 0.80 | 0.05 | 0.66 | 0.04 | 0.64 | 0.02 | 0.79 | 0.10 | 0.88 | 0.02 |
| | | DINO-Patch | 0.56 | 0.04 | 0.74 | 0.06 | 0.78 | 0.04 | 0.57 | 0.06 | 0.76 | 0.11 | 0.92 | 0.03 |
| | | LDM | 0.67 | 0.08 | 0.75 | 0.08 | 0.67 | 0.11 | 0.61 | 0.04 | 0.76 | 0.11 | 0.91 | 0.05 |
| | | LRDM | 0.67 | 0.08 | 0.75 | 0.08 | 0.67 | 0.11 | 0.61 | 0.04 | 0.76 | 0.11 | 0.91 | 0.05 |
| | | PredNet | 0.72 | 0.06 | 0.77 | 0.06 | 0.63 | 0.11 | 0.81 | 0.06 | 0.90 | 0.09 | 0.69 | 0.13 |
| | | RES-KNN | 0.65 | 0.09 | 0.76 | 0.08 | 0.69 | 0.07 | 0.69 | 0.09 | 0.83 | 0.12 | 0.88 | 0.07 |
| | | RSSM-PPM | 0.65 | 0.02 | 0.76 | 0.06 | 0.69 | 0.03 | 0.63 | 0.02 | 0.77 | 0.09 | 0.85 | 0.03 |
| | tiny | AE-Pred | 0.59 | 0.01 | 0.66 | 0.02 | 0.85 | 0.02 | 0.56 | 0.01 | 0.51 | 0.02 | 0.93 | 0.01 |
| | | CLIP-Sim | 0.72 | 0.01 | 0.72 | 0.02 | 0.63 | 0.02 | 0.61 | 0.01 | 0.54 | 0.02 | 0.90 | 0.02 |
| | | DINO-Patch | 0.51 | 0.02 | 0.62 | 0.02 | 0.78 | 0.03 | 0.50 | 0.01 | 0.47 | 0.02 | 0.96 | 0.01 |
| | | LDM | 0.55 | 0.02 | 0.64 | 0.01 | 0.85 | 0.02 | 0.53 | 0.01 | 0.48 | 0.02 | 0.96 | 0.02 |
| | | LRDM | 0.55 | 0.02 | 0.64 | 0.01 | 0.85 | 0.02 | 0.53 | 0.01 | 0.48 | 0.02 | 0.96 | 0.02 |
| | | PredNet | 0.63 | 0.02 | 0.68 | 0.02 | 0.81 | 0.02 | 0.65 | 0.02 | 0.61 | 0.03 | 0.92 | 0.02 |
| | | RES-KNN | 0.50 | 0.02 | 0.61 | 0.02 | 0.76 | 0.03 | 0.53 | 0.03 | 0.50 | 0.03 | 0.96 | 0.01 |
| | | RSSM-PPM | 0.64 | 0.02 | 0.67 | 0.02 | 0.67 | 0.03 | 0.61 | 0.02 | 0.55 | 0.03 | 0.90 | 0.02 |
| SAP-Goal1 | strong | AE-Pred | 0.81 | 0.06 | 0.82 | 0.07 | 0.43 | 0.08 | 0.67 | 0.04 | 0.85 | 0.07 | 0.91 | 0.05 |
| | | CLIP-Sim | 0.65 | 0.02 | 0.78 | 0.06 | 0.75 | 0.04 | 0.60 | 0.03 | 0.79 | 0.06 | 0.92 | 0.02 |
| | | DINO-Patch | 0.53 | 0.03 | 0.77 | 0.05 | 0.77 | 0.03 | 0.55 | 0.03 | 0.76 | 0.06 | 0.92 | 0.04 |
| | | LDM | 0.82 | 0.06 | 0.83 | 0.07 | 0.41 | 0.08 | 0.67 | 0.03 | 0.85 | 0.07 | 0.91 | 0.04 |
| | | LRDM | 0.82 | 0.06 | 0.83 | 0.07 | 0.41 | 0.08 | 0.67 | 0.03 | 0.85 | 0.07 | 0.91 | 0.04 |
| | | PredNet | 0.80 | 0.06 | 0.82 | 0.07 | 0.45 | 0.08 | 0.66 | 0.04 | 0.85 | 0.07 | 0.92 | 0.04 |
| | | RES-KNN | 0.56 | 0.02 | 0.75 | 0.06 | 0.74 | 0.03 | 0.54 | 0.03 | 0.76 | 0.07 | 0.96 | 0.01 |
| | | RSSM-PPM | 0.61 | 0.03 | 0.77 | 0.06 | 0.74 | 0.03 | 0.62 | 0.03 | 0.78 | 0.08 | 0.90 | 0.03 |
| | tiny | AE-Pred | 0.73 | 0.06 | 0.72 | 0.03 | 0.54 | 0.08 | 0.59 | 0.05 | 0.63 | 0.08 | 0.94 | 0.03 |
| | | CLIP-Sim | 0.64 | 0.02 | 0.70 | 0.03 | 0.74 | 0.02 | 0.59 | 0.03 | 0.61 | 0.07 | 0.93 | 0.02 |
| | | DINO-Patch | 0.55 | 0.03 | 0.68 | 0.01 | 0.75 | 0.02 | 0.56 | 0.01 | 0.57 | 0.05 | 0.91 | 0.02 |
| | | LDM | 0.74 | 0.07 | 0.74 | 0.03 | 0.52 | 0.09 | 0.59 | 0.05 | 0.63 | 0.07 | 0.94 | 0.03 |
| | | LRDM | 0.74 | 0.07 | 0.74 | 0.03 | 0.52 | 0.09 | 0.59 | 0.05 | 0.63 | 0.07 | 0.94 | 0.03 |
| | | PredNet | 0.72 | 0.06 | 0.72 | 0.03 | 0.56 | 0.08 | 0.59 | 0.05 | 0.64 | 0.08 | 0.94 | 0.03 |
| | | RES-KNN | 0.57 | 0.03 | 0.67 | 0.02 | 0.73 | 0.03 | 0.55 | 0.01 | 0.57 | 0.06 | 0.96 | 0.01 |
| | | RSSM-PPM | 0.59 | 0.02 | 0.68 | 0.02 | 0.75 | 0.03 | 0.58 | 0.02 | 0.58 | 0.06 | 0.91 | 0.03 |
| SAP-Goal2 | strong | AE-Pred | 0.85 | 0.04 | 0.84 | 0.05 | 0.38 | 0.05 | 0.64 | 0.03 | 0.78 | 0.07 | 0.89 | 0.02 |
| | | CLIP-Sim | 0.57 | 0.02 | 0.72 | 0.05 | 0.79 | 0.03 | 0.57 | 0.01 | 0.71 | 0.07 | 0.93 | 0.01 |
| | | DINO-Patch | 0.47 | 0.04 | 0.67 | 0.05 | 0.80 | 0.03 | 0.51 | 0.04 | 0.69 | 0.06 | 0.95 | 0.02 |

Table 14: Detailed results on image observations for all detectors and environments (continued).

| env | strength | detector | AUROC mean | AUROC std | AUPR mean | AUPR std | FPR95 mean | FPR95 std | g. AUROC mean | g. AUROC std | g. AUPR mean | g. AUPR std | g. FPR95 mean | g. FPR95 std |
|---|---|---|---|---|---|---|---|---|---|---|---|---|---|---|
| URM-PnP | tiny | LDM | 0.83 | 0.05 | 0.81 | 0.06 | 0.36 | 0.06 | 0.65 | 0.03 | 0.80 | 0.07 | 0.90 | 0.02 |
| | | LRDM | 0.81 | 0.05 | 0.81 | 0.06 | 0.40 | 0.05 | 0.61 | 0.02 | 0.74 | 0.09 | 0.89 | 0.02 |
| | | PredNet | 0.76 | 0.06 | 0.78 | 0.07 | 0.50 | 0.06 | 0.60 | 0.03 | 0.76 | 0.08 | 0.90 | 0.01 |
| | | RES-KNN | 0.52 | 0.02 | 0.68 | 0.06 | 0.75 | 0.03 | 0.50 | 0.01 | 0.66 | 0.07 | 0.94 | 0.02 |
| | | RSSM-PPM | 0.50 | 0.04 | 0.68 | 0.06 | 0.84 | 0.04 | 0.47 | 0.05 | 0.65 | 0.09 | 0.96 | 0.02 |
| | | AE-Pred | 0.79 | 0.08 | 0.76 | 0.05 | 0.47 | 0.09 | 0.60 | 0.05 | 0.57 | 0.06 | 0.90 | 0.03 |
| | | CLIP-Sim | 0.58 | 0.03 | 0.65 | 0.02 | 0.77 | 0.04 | 0.56 | 0.02 | 0.51 | 0.04 | 0.94 | 0.01 |
| | | DINO-Patch | 0.47 | 0.05 | 0.58 | 0.03 | 0.79 | 0.03 | 0.49 | 0.04 | 0.46 | 0.04 | 0.95 | 0.03 |
| | strong | LDM | 0.76 | 0.07 | 0.72 | 0.04 | 0.47 | 0.10 | 0.60 | 0.05 | 0.58 | 0.06 | 0.92 | 0.02 |
| | | LRDM | 0.74 | 0.07 | 0.72 | 0.04 | 0.49 | 0.09 | 0.57 | 0.04 | 0.51 | 0.05 | 0.90 | 0.03 |
| | | PredNet | 0.69 | 0.06 | 0.67 | 0.03 | 0.58 | 0.07 | 0.56 | 0.04 | 0.53 | 0.05 | 0.92 | 0.03 |
| | | RES-KNN | 0.52 | 0.04 | 0.59 | 0.03 | 0.73 | 0.04 | 0.50 | 0.04 | 0.46 | 0.05 | 0.95 | 0.03 |
| | | RSSM-PPM | 0.51 | 0.02 | 0.58 | 0.02 | 0.81 | 0.01 | 0.49 | 0.02 | 0.45 | 0.03 | 0.95 | 0.02 |
| | | AE-Pred | 0.48 | 0.06 | 0.53 | 0.03 | 0.68 | 0.09 | 0.45 | 0.04 | 0.49 | 0.03 | 0.95 | 0.03 |
| | | CLIP-Sim | 0.56 | 0.07 | 0.57 | 0.05 | 0.81 | 0.04 | 0.55 | 0.06 | 0.58 | 0.05 | 0.94 | 0.04 |
| | | DINO-Patch | 0.52 | 0.03 | 0.55 | 0.03 | 0.83 | 0.03 | 0.53 | 0.04 | 0.55 | 0.03 | 0.95 | 0.02 |
| URM-Reach | tiny | LDM | 0.50 | 0.05 | 0.55 | 0.03 | 0.70 | 0.07 | 0.49 | 0.04 | 0.51 | 0.03 | 0.92 | 0.05 |
| | | LRDM | 0.50 | 0.05 | 0.55 | 0.03 | 0.70 | 0.07 | 0.49 | 0.04 | 0.51 | 0.03 | 0.92 | 0.05 |
| | | PredNet | 0.29 | 0.19 | 0.45 | 0.11 | 0.86 | 0.14 | 0.40 | 0.16 | 0.45 | 0.12 | 0.92 | 0.15 |
| | | RES-KNN | 0.46 | 0.09 | 0.52 | 0.07 | 0.84 | 0.06 | 0.49 | 0.09 | 0.53 | 0.08 | 0.93 | 0.05 |
| | | RSSM-PPM | 0.43 | 0.06 | 0.49 | 0.03 | 0.94 | 0.02 | 0.49 | 0.06 | 0.50 | 0.04 | 0.94 | 0.04 |
| | | AE-Pred | 0.47 | 0.02 | 0.52 | 0.02 | 0.66 | 0.05 | 0.43 | 0.01 | 0.47 | 0.01 | 0.96 | 0.02 |
| | | CLIP-Sim | 0.46 | 0.03 | 0.52 | 0.02 | 0.83 | 0.03 | 0.46 | 0.03 | 0.51 | 0.03 | 0.98 | 0.01 |
| | | DINO-Patch | 0.49 | 0.01 | 0.53 | 0.01 | 0.82 | 0.03 | 0.49 | 0.02 | 0.52 | 0.01 | 0.96 | 0.02 |
| | strong | LDM | 0.47 | 0.02 | 0.54 | 0.02 | 0.69 | 0.04 | 0.45 | 0.02 | 0.49 | 0.02 | 0.96 | 0.03 |
| | | LRDM | 0.47 | 0.02 | 0.54 | 0.02 | 0.69 | 0.04 | 0.45 | 0.02 | 0.49 | 0.02 | 0.96 | 0.03 |
| | | PredNet | 0.20 | 0.12 | 0.41 | 0.06 | 0.92 | 0.07 | 0.30 | 0.10 | 0.40 | 0.07 | 0.98 | 0.02 |
| | | RES-KNN | 0.36 | 0.04 | 0.47 | 0.03 | 0.86 | 0.02 | 0.39 | 0.04 | 0.44 | 0.03 | 0.96 | 0.02 |
| | | RSSM-PPM | 0.39 | 0.03 | 0.47 | 0.01 | 0.95 | 0.01 | 0.43 | 0.04 | 0.47 | 0.03 | 0.96 | 0.01 |
| | | AE-Pred | 0.61 | 0.02 | 0.62 | 0.03 | 0.70 | 0.03 | 0.56 | 0.03 | 0.56 | 0.03 | 0.93 | 0.03 |
| | | CLIP-Sim | 0.59 | 0.06 | 0.60 | 0.04 | 0.83 | 0.06 | 0.60 | 0.06 | 0.60 | 0.06 | 0.90 | 0.05 |
| | | DINO-Patch | 0.60 | 0.06 | 0.60 | 0.04 | 0.79 | 0.07 | 0.61 | 0.06 | 0.60 | 0.07 | 0.87 | 0.05 |
| URR-Reach | tiny | LDM | 0.60 | 0.03 | 0.62 | 0.02 | 0.72 | 0.04 | 0.56 | 0.03 | 0.57 | 0.03 | 0.93 | 0.01 |
| | | LRDM | 0.61 | 0.03 | 0.62 | 0.02 | 0.68 | 0.04 | 0.57 | 0.03 | 0.56 | 0.03 | 0.92 | 0.03 |
| | | PredNet | 0.50 | 0.09 | 0.55 | 0.05 | 0.78 | 0.07 | 0.57 | 0.08 | 0.55 | 0.07 | 0.80 | 0.10 |
| | | RES-KNN | 0.60 | 0.08 | 0.61 | 0.05 | 0.79 | 0.09 | 0.63 | 0.08 | 0.62 | 0.08 | 0.85 | 0.10 |
| | | RSSM-PPM | 0.53 | 0.02 | 0.55 | 0.02 | 0.87 | 0.01 | 0.53 | 0.03 | 0.53 | 0.03 | 0.93 | 0.01 |
| | | AE-Pred | 0.60 | 0.01 | 0.62 | 0.03 | 0.70 | 0.01 | 0.55 | 0.02 | 0.55 | 0.03 | 0.94 | 0.02 |
| | | CLIP-Sim | 0.53 | 0.04 | 0.57 | 0.02 | 0.88 | 0.04 | 0.53 | 0.04 | 0.54 | 0.03 | 0.94 | 0.04 |
| | | DINO-Patch | 0.54 | 0.03 | 0.57 | 0.02 | 0.84 | 0.04 | 0.55 | 0.03 | 0.55 | 0.03 | 0.92 | 0.04 |
| | strong | LDM | 0.59 | 0.02 | 0.62 | 0.02 | 0.73 | 0.02 | 0.55 | 0.02 | 0.56 | 0.02 | 0.94 | 0.01 |
| | | LRDM | 0.60 | 0.02 | 0.61 | 0.02 | 0.70 | 0.02 | 0.56 | 0.02 | 0.55 | 0.02 | 0.93 | 0.02 |
| | | PredNet | 0.45 | 0.07 | 0.53 | 0.04 | 0.82 | 0.06 | 0.52 | 0.06 | 0.52 | 0.06 | 0.88 | 0.07 |
| | | RES-KNN | 0.51 | 0.06 | 0.56 | 0.03 | 0.85 | 0.06 | 0.54 | 0.05 | 0.55 | 0.04 | 0.92 | 0.06 |
| | | RSSM-PPM | 0.52 | 0.01 | 0.55 | 0.02 | 0.88 | 0.01 | 0.52 | 0.02 | 0.53 | 0.03 | 0.93 | 0.01 |
| | | AE-Pred | 0.60 | 0.05 | 0.60 | 0.04 | 0.66 | 0.06 | 0.55 | 0.06 | 0.52 | 0.07 | 0.92 | 0.04 |
| | | CLIP-Sim | 0.58 | 0.04 | 0.56 | 0.02 | 0.83 | 0.03 | 0.56 | 0.05 | 0.54 | 0.04 | 0.93 | 0.03 |
| | | DINO-Patch | 0.55 | 0.04 | 0.55 | 0.03 | 0.83 | 0.03 | 0.52 | 0.03 | 0.50 | 0.02 | 0.94 | 0.02 |
| URR-PnP | tiny | LDM | 0.64 | 0.05 | 0.63 | 0.04 | 0.64 | 0.08 | 0.60 | 0.07 | 0.56 | 0.08 | 0.87 | 0.08 |
| | | LRDM | 0.64 | 0.05 | 0.63 | 0.04 | 0.64 | 0.08 | 0.60 | 0.07 | 0.56 | 0.08 | 0.87 | 0.08 |
| | | PredNet | 0.43 | 0.03 | 0.48 | 0.02 | 0.84 | 0.02 | 0.47 | 0.03 | 0.46 | 0.02 | 0.95 | 0.03 |
| | | RES-KNN | 0.62 | 0.03 | 0.59 | 0.03 | 0.72 | 0.05 | 0.58 | 0.05 | 0.56 | 0.05 | 0.89 | 0.04 |
| | | RSSM-PPM | 0.56 | 0.04 | 0.52 | 0.02 | 0.84 | 0.04 | 0.55 | 0.05 | 0.51 | 0.04 | 0.90 | 0.04 |
| | | AE-Pred | 0.58 | 0.03 | 0.59 | 0.03 | 0.67 | 0.03 | 0.53 | 0.02 | 0.50 | 0.01 | 0.94 | 0.02 |
| | | CLIP-Sim | 0.60 | 0.02 | 0.56 | 0.02 | 0.83 | 0.02 | 0.56 | 0.03 | 0.53 | 0.03 | 0.94 | 0.02 |
| | | DINO-Patch | 0.55 | 0.03 | 0.54 | 0.02 | 0.83 | 0.04 | 0.52 | 0.03 | 0.50 | 0.02 | 0.93 | 0.01 |
| | strong | LDM | 0.59 | 0.05 | 0.59 | 0.04 | 0.69 | 0.06 | 0.54 | 0.05 | 0.51 | 0.04 | 0.90 | 0.04 |
| | | LRDM | 0.59 | 0.05 | 0.59 | 0.04 | 0.69 | 0.06 | 0.54 | 0.05 | 0.51 | 0.04 | 0.90 | 0.04 |
| | | PredNet | 0.44 | 0.03 | 0.49 | 0.01 | 0.82 | 0.01 | 0.46 | 0.03 | 0.46 | 0.02 | 0.96 | 0.03 |
| | | RES-KNN | 0.62 | 0.03 | 0.59 | 0.01 | 0.72 | 0.04 | 0.56 | 0.03 | 0.54 | 0.03 | 0.90 | 0.03 |
| | | RSSM-PPM | 0.56 | 0.03 | 0.53 | 0.01 | 0.85 | 0.02 | 0.54 | 0.03 | 0.51 | 0.02 | 0.91 | 0.03 |

