# OpenReview forum: "Anomaly-Gym: A Benchmark for Anomaly Detection in Reinforcement Learning Environments"
_TMLR — Under review for TMLR_

### Review · Reviewer_spib · 2026-04-14

**Summary Of Contributions:**

**Summary:** This paper argues that anomaly detection (AD) in RL is important for deployment, but current works are fragmented and typically evaluated on small, self-chosen settings. To address this, the authors first formalize anomaly detection in RL as detecting deviations from the nominal deployment distribution induced by a given fixed policy, and then build a benchmark around that formulation. The benchmark, named Anomaly-Gym, covers observation, action, and dynamics anomalies, and calibrates anomaly strength levels using normalized policy degradation. The authors conduct a set of experiments, which evaluate a range of classical and learned detectors on both vector and image observations, and analyze aggregate detection quality as well as timing metrics. The authors provide some takeaway observations: simple methods can work really well, dynamics-aware methods are better for action/dynamics anomalies, and image-based detection is still challenging.

**Strength:**
- The problem is well-motivated and important for safe RL deployment. I appreciate the explicit desiderata (ED1–ED6, AD1–AD5), which provide a transparent framework for design decisions and give future benchmark builders a template to critique or extend. Overall, the paper is well written and very readable.
- The benchmark is broad, which covers 10 environments, 4 domains, 3 anomaly classes, 4 strength levels, and 2 observation modalities.
- The observations are interesting. They demonstrate that the simple method works well for simple anomalies, even better than the complex ones, while there is still room for further research in harder anomalies.
- Almost all claims in the papers are supported.

**Weakness:**  Overall, I think this is good work that could pave the way for future work on AD in RL settings. However, real-world validation is limited to a single URR-Reach task, and the sim-to-real claim is explicitly qualitative. Adding more real-world tasks could significantly strengthen the paper's contribution.

**Additional Comments:**

No.

**Audience:**

Yes

**Audience Explanation:**

This paper will be of interest to researchers working on topics of anomaly detection in RL.

**Broader Impact Concerns:**

There are no broader impact concerns.

**Claims And Evidence:**

No

**Claims Explanation:**

There is one claim that is partially supported. The paper claims that trends in the real-world URR-Reach task qualitatively match the simulated URM-Reach task. However, just one environment is a narrow basis for a sim-to-real claim, and validation on more real tasks is still needed.

**Requested Changes:**

- Could you add a compact table summarizing all included and excluded anomaly-environment-strength combinations, with reasons for exclusion?
- Could you update the link to the repository? The current link seems expired.
- Could you fix the presentation of Figure 2 in the main paper. Besides that, tables 12–13 are very dense and could use a brief reading guide.

---

> ### Author Response · Authors · 2026-04-23
> **Answer to Reviewer spib**
>
> We thank the reviewer for their valuable feedback, which will strengthen the final version of the paper.
>
> In line with TMLR’s recommendation, we will update the manuscript once all reviews have been submitted. In the revised version, we will make the following changes:
> - We will add a Table with an overview of all omitted environment-anomaly-strength combinations.
> - We will add a caption/description for Figure 2.
> - We will add a reading guide for the detailed results tables.
>
> Regarding the link to the repository: We double checked and the link seems to work on our side. If the issue persists, we would be grateful for any additional details that could help us diagnose it.

---

> > ### Comment · Reviewer_spib · 2026-04-28
> >
> > Thank you for your rebuttal. I will read the paper once the revision is updated. In the meantime, I can now see the GitHub repo.

---

> > > ### Author Response · Authors · 2026-07-19
> > > **Updated Manuscript**
> > >
> > > We thank the reviewer for their patience. As all reviews have now been made available, we updated the manuscript:
> > > - Overview of omitted environments (Table 9)
> > > - Updated description (Figure 2)
> > > - Reading guide for detailed results (A.11)

---

### Review · Reviewer_36yb · 2026-06-23

**Summary Of Contributions:**

The paper introduces Anomaly-Gym, a comprehensive benchmark for anomaly detection in reinforcement learning settings. It provides a principled evaluation suite covering 10 environments, 25 anomaly types, 4 strength levels, and multiple sensor modalities across simulated and real-world tasks. Through extensive experiments, it shows the strengths and limitations of existing methods, revealing that simple methods can work well for observational disturbances while more complex approaches are needed for action or dynamics perturbations.

**Audience:**

Yes

**Audience Explanation:**

The benchmark and analysis benefit the TMLR audience within the field of RL.

**Broader Impact Concerns:**

The work has clear positive potential for improving the safety of RL agents in real-world and safety-critical deployments, but it also raises several broader-impact considerations. In particular, anomaly detection failures could lead to unsafe behavior if anomalies are missed, while excessive false alarms could cause unnecessary system shutdowns or reduced trust in autonomous systems. Since the benchmark includes real-world tasks and multiple sensor modalities, the authors should clarify whether privacy, data collection, and potential misuse risks are adequately addressed. A Broader Impact Statement should also discuss how benchmark results should be interpreted cautiously to avoid overstating the reliability of anomaly detection methods in high-stakes environments.

**Claims And Evidence:**

Yes

**Claims Explanation:**

Yes the claims made in the submission supported by the experimental results and analysis.

**Requested Changes:**

1. Regarding the anomalies considered, it would be beneficial to include more diverse perturbations, such as occlusion. The authors may refer to [a] and incorporate additional perturbation types into the experiments.

[a] Kumar, Vishesh, Shivam Shukla, and Akshay Agarwal. "Robustness Benchmarking of Convolutional and Transformer Architectures for Image Classification." IEEE Transactions on Big Data (2025).

2. According to Section 6.1, most of the adopted baselines are not very recent. The authors are encouraged to include more recent RL approaches in the experiments, such as methods proposed in 2025 and 2026.

3. In Figure 3, some bars in [e] and [f] exceed the boundaries of the figure. The authors are encouraged to revise the figure accordingly.

4. The paper lacks qualitative results. The authors are suggested to provide more qualitative analyses to offer a more fine-grained understanding of the effectiveness of different approaches. Additional failure cases and t-SNE visualizations could further enrich the discussion.

5. The paper does not compare the efficiency of different RL approaches when applied to anomaly data. The authors are encouraged to include an efficiency comparison to better assess the practical applicability of the evaluated methods.

---

> ### Author Response · Authors · 2026-06-28
> **Answer to Reviewer 36yb**
>
> We thank the reviewer for their detailed review and constructive suggestions. Below, we address the requested changes point by point.
>
> ### 1. Additional anomaly types, including occlusion.
>
> We agree that incorporating a broader range of perturbations, including visual occlusions, is an interesting direction. Our current benchmark already covers 12 anomaly types across all 10 environments, resulting in a diverse set of perturbations with systematically controlled anomaly strengths. In contrast to the perturbations considered in our benchmark, occlusions are difficult to parameterize along a comparable continuous severity scale, since their effect depends strongly on object placement, size, shape, and scene context. We therefore believe that visual perturbations such as occlusions deserve a dedicated study with a carefully designed protocol. We will clarify this design choice in the paper and discuss occlusion-based anomalies as an important direction for future work.
>
> ### 2. Inclusion of more recent RL or anomaly-detection baselines.
>
> We thank the reviewer for this suggestion. We would like to clarify that our benchmark includes recent AD methods for RL, including RSSM-PPM from 2025. To the best of our knowledge, this is the most recent method specifically proposed for AD in RL settings. We kindly ask the reviewer to directly point us to any methods that we missed in our evaluation.
>
> If the comment instead refers to RL algorithms for generating rollout data, we note that we already compare different rollout policies in Section 6.3, “Influence of rollout policy,” and report the results in Table 5. This analysis evaluates the effect of different RL policies on anomaly-detection performance and on performance degradation under perturbations.
>
> ### 3. Figure 3 visualization.
> We thank the reviewer for pointing this out. Although some bars in panels (e) and (f) may visually appear to exceed the plot boundaries, all AUROC values are below 1.0, which is the theoretical maximum. We will revise the figure formatting to avoid this misleading visual impression.
>
> ### 4. Qualitative analysis.
> We agree with the reviewer that additional qualitative results would strengthen the paper. In the revised version, we will add qualitative analyses of detection scores over time for individual episodes. This will provide a more fine-grained view of how different methods behave under specific anomaly types and help illustrate both successful detections and failure cases. We believe this addition will make the empirical comparison more interpretable and will complement the aggregate quantitative results.
>
> ### 5. Efficiency comparison of RL approaches.
> We thank the reviewer for raising this point. We would appreciate clarification regarding the intended notion of efficiency: computational efficiency, sample efficiency, or robustness-related efficiency under perturbations. In the current manuscript, Section 6.3, “Influence of rollout policy,” and Table 5 already compare different RL algorithms with respect to their performance degradation under perturbations.
>
> ### Broader Impact Statement
> We thank the reviewer for raising these important broader-impact considerations. We agree that benchmark results should be interpreted cautiously and should not be taken as direct evidence of deployment readiness in safety-critical settings. In the revised manuscript, we will briefly clarify these limitations in the Broader Impact Statement, including the potential consequences of missed anomalies, false alarms, and over-reliance on benchmark performance. We will also state that our dataset does not involve personally identifiable information or human-subject data. Finally, we will note that deployment in high-stakes settings requires additional validation and safety mechanisms beyond the benchmark evaluation.

---

> > ### Author Response · Authors · 2026-07-19
> > **Updated Manuscript**
> >
> > As all reviews have now been made available, we updated the manuscript with the following points:
> >
> > - We clarified the motivation for the selection of anomaly types (section 5.2) and added a discussion on additional visual perturbations (section 7).
> > - We updated figure 3.
> > - We added additional qualitative analysis (section 6.3 Analysis of Detection Timing, figure 4)
> > - We added an impact statement.
> >
> > We remain happy to continue the discussion on any outstanding points.

---

### Review · Reviewer_cahQ · 2026-07-15

**Summary Of Contributions:**

The paper presents a new benchmark discovering on-policy anomalies in RL environments with external predictors. The benchmark consists of ten environments with broad anomaly types in different levels of intensity. It is argued that prior benchmarks largely focus on constrained settings such as discrete action spaces or simple state spaces. The integrated environments comprise MuJoCo and UR variant, Carla Lanekeep, URR or SAP. Possible anomalies on the state space include perturbations, drifts, temporal noise and others. The authors also impose/define meaningful desiderata for their benchmark such as diversity or realism, which the benchmark fulfills.

The authors provide an evaluation of the benchmark with several anomaly detectors, including one- class SVM or isolation forest with established metrics. Lastly, the paper includes sensible discussion on limitations of the approach.

**Audience:**

Yes

**Audience Explanation:**

As stated, the problem is very important even though it is not tackling policy robustness. One can infer important decisions out of knowing that the dynamics model changed which increase RL safety. So it is for sure relevant to TMLR.

**Broader Impact Concerns:**

No.

**Claims And Evidence:**

Yes

**Claims Explanation:**

The benchmark tackles an important problem. While it clearly separates the problem from estimating robustness of the RL policy, it can detect if the environment changes significantly. While the policy might or might not generalize, this information is very useful to detect if sensors or actuators changed or if the complete dynamics model is starting to diverge.

The paper sufficiently motivates the novelty and usefulness of the benchmark. Related works do not sufficiently cover continuous state action spaces. Robotic tasks are increasingly relevant and sufficiently complex, so using them as part of the benchmark is a good approach.

The experiments use a good selection of baseline approaches and metrics for anomaly detection.

**Requested Changes:**

- Could the approach also handle multiple agents / MDP extensions to multiple agents? These become increasingly relevant, so it would be interesting to discuss
-  Does the paper include representative sets of revent anomaly detectors? The choice of algorithms in the evaluation is not fully motivated.

---

> ### Author Response · Authors · 2026-07-19
> **Answer to Reviewer cahQ**
>
> We thank the reviewer for their constructive feedback. Below we address each points in turn.
>
> 1. We thank the reviewer for raising this point. We agree that multi-agent settings are increasingly relevant. Our current benchmark focuses on single-agent RL environments, but the underlying evaluation idea could be extended by defining nominal behavior over the joint interaction distribution induced by multiple agents. However, such an extension adds additional benchmark-design challenges, including non-stationarity and anomaly attribution across agents and the environment. We will add this discussion to Section 7 - Outlook (iii)  and identify multi-agent anomaly detection as an important future extension.
>
> 2. We thank the reviewer for pointing this out. We agree that the motivation for the detector selection should be made clearer. Our evaluation is intended to cover representative detector families rather than provide an exhaustive leaderboard. Specifically, we include classical one-class / nearest-neighbor baselines, RL-specific sequence and dynamics-modeling methods, and image-based reconstruction, prediction, and representation-based methods. This selection reflects the main methodological families applicable to external, policy-agnostic AD from nominal rollouts. We will clarify these selection criteria at the beginning of Section 6.1.
>
> We will update the manuscript with the above points.